MPP-2022-11

# Looking for structure in the cobordism conjecture

David Andriot[1], Nils Carqueville[2] and Niccolò Cribiori[3]

[1]*Laboratoire d'Annecy-le-Vieux de Physique Théorique (LAPTh),*
*UMR 5108, CNRS, Université Savoie Mont Blanc (USMB),*
*9 Chemin de Bellevue, 74940 Annecy, France*

[2]*Universität Wien, Fakultät für Physik, Boltzmanngasse 5, 1090 Wien, Österreich*

[3]*Max-Planck-Institut für Physik (Werner-Heisenberg-Institut)*
*Föhringer Ring 6, 80805 München, Germany*

andriot@lapth.cnrs.fr; nils.carqueville@univie.ac.at; cribiori@mpp.mpg.de

**Abstract**

The cobordism conjecture of the swampland program states that the bordism group of quantum gravity must be trivial. We investigate this statement in several directions, on both the mathematical and physical side. We consider the Whitehead tower construction as a possible organising principle for the topological structures entering the formulation of the conjecture. We discuss why and how to include geometric structures in bordism groups, such as higher U(1)-bundles with connection. The inclusion of magnetic defects is also addressed in some detail. We further elaborate on how the conjecture could predict Kaluza–Klein monopoles, and we study the gravity decoupling limit in the cobordism conjecture, with a few observations on NSNS string backgrounds. We end with comments in relation to T-duality, as well as the finiteness conjecture.

# 1 Introduction

The swampland program [1–3] aims at characterising which low-energy effective theories are consistent with quantum gravity. For this purpose, several criteria have been conjectured and tested in recent years. They typically encode information on one or more specific properties of quantum gravity, which a priori are independent of one another. Nevertheless, it has been realised that several of these conjectures are indeed related, sometimes in a non-trivial and unexpected manner, thus forming an interesting "web" of conjectures. This fact is expected to hint at the existence of deeper principles characterising quantum gravity completely.

In this paper, we concentrate on one of the central conjectures in this web, the so-called cobordism conjecture, proposed in [4]. We investigate several of its aspects, from both the mathematical and physical side. Indeed, given the speculative nature of some of the swampland conjectures, one might feel that they are neither concrete nor sound on the more formal side. One of the aims of the present work is to contribute to the integration of concreteness, generality, and rigour in this context. Accordingly, we will give further evidence that the cobordism conjecture is naturally related to deep and useful mathematical structures, some of which still call for a physical interpretation.

The swampland cobordism conjecture is about quantum gravity backgrounds in the form of a $(D + k)$-dimensional spacetime, with $D$ external directions (including time) and a $k$-dimensional compact manifold $M$ (which may carry extra structure such as an orientation, a Riemannian metric or a $G$-bundle). These backgrounds appear naturally in string compactifications, but in the spirit of the swampland program one can also follow a genuinely bottom-up strategy and concentrate on the effective theory living in $D$ dimensions, and the conjecture can a priori be considered independently of string theory. One of the main motivations behind the conjecture is to give a mathematically rigorous formulation to the common expectation that quantum gravity is unique, in the sense that all quantum gravity compactifications and resulting $D$-dimensional effective theories enjoy a certain equivalence relation. In string theory, it is natural to have relations between backgrounds, such as dualities, singularity resolutions, geometric transitions and related phenomena. The equivalence relation proposed in [4] is meant to go further. It turns out that the mathematical language best suited to this purpose may be that of bordisms[1] and bordism groups (reviewed in Section 2.1). The conjecture postulates the existence of backgrounds (and thus of compact $k$-dimensional manifolds) with a (currently undetermined) "quantum gravity structure" QG, such that the corresponding bordism group is trivial, $\Omega_k^{\text{QG}} = 0$. Given that bordism is an equivalence relation and that the bordism group is made up of equivalence classes, the conjecture states that in quantum gravity there is only one single equivalence class, which is the trivial and only element of the group. All of the consistent QG backgrounds are then representatives of this class. In practice, this means that (the disjoint union of) any two compact $k$-dimensional manifolds, together with the extra structure which they have to carry to be relevant in quantum gravity, always form the boundary of a $(k + 1)$-dimensional manifold, to which the aforementioned extra structure is extended. Such manifolds are called bordisms, and, crucially, they can allow for topology changes between the two original $k$-dimensional manifolds. In [4], bordisms with QG-structure are physically interpreted as a certain dynamically allowed process connecting them and under which the physics in the $D$-dimensional effective theory is not necessarily preserved. Mathematically, the cobordism conjecture can be seen as a necessary condition on the unknown structure QG; "solving" the equations $\Omega_k^{\text{QG}} = 0$ (where $k$ may take all allowed values) for QG may give insights as to which geometric or "quantum" structures backgrounds allowed in quantum gravity must have.

Establishing whether or not any two given compact manifolds are bordant is non-trivial already at the mathematical level. Therefore, the existence of a process connecting any two quantum gravity backgrounds is far from obvious too. Indeed, even if the conjecture is quite recent, already a number of works appeared studying some of its aspects and consequences, including [5–11]. Bordisms are relevant in several areas in mathematics and physics, a non-exhaustive list of related works is [12–16] for anomalies, [17, 18] in relation to bubbles of nothing, [19] for the conjecture in a holographic context, [20] about "Hypothesis H" on charge quantisation in M-theory. An introduction to some of the mathematical material, in a physical context, can be found e. g. in [21, 22], that appeared prior to [4].

Identifying the unknown QG-structure entering the formulation of the conjecture would clearly present a major step forward. This is a challenging task, if not out of reach at present, given that it would imply a much deeper understanding of quantum gravity itself. In the paper [4], a concrete recipe was proposed to get closer to QG. Suppose we start with an

---

[1]We use the term "bordism" rather than "cobordism" because it is shorter, with the exception of the conjecture's name. Both words are commonly used in the literature to describe the same concept.

"approximative QG-structure" $\widetilde{\text{QG}}$ such that the associated bordism group is non-trivial, $\Omega_k^{\widetilde{\text{QG}}} \neq 0$. One should then refine the structure $\widetilde{\text{QG}}$ to another structure QG in such a way that the bordism group with the new structure is indeed trivial. Such a strategy is successful in the sense that it led the authors of [4] to recover known objects (defects) in string theory, such as branes, and to even predict new ones; the structures encountered in [4] are topological tangential structures like orientations, spin, or string structures.

In the present paper, we first build on the above idea to "refine" spacetime structures and propose to exploit a mathematical construction known as the Whitehead tower as an organising principle governing the refinements of the various approximative $\widetilde{\text{QG}}$-structures (see Section 3). Roughly speaking, the Whitehead tower arranges topological spaces according to their degree of connectedness. The construction is such that when passing from one level to the next one, there is a lift of the given tangential structure, which we interpret as the mathematically precise definition behind the aforementioned refinement. Interestingly, even if it might still end up being non-trivial, we observe that the bordism groups become systematically smaller when climbing Whitehead towers of spaces that are related to structures which naturally appear in string theory. Thus, we believe that Whitehead towers may be of some help in the identification of the unknown QG-structure.

Since any realistic physical setup contains gauge fields and fluxes (mathematically described by connections of higher principal bundles), we extend the original formulation of the cobordism conjecture in order to include them as well as "geometric structures" in general (see Section 4). This step requires some additional mathematical concepts, such as simplicial presheaves and twisted differential cohomology. We give a definition of bordism groups for any fixed geometric structure, and we observe that in the case of connections including them does not make a difference in the end. Beyond such examples that form contractible spaces, we are not aware of any explicit computations of bordism groups with geometric structure, but we believe that they should be investigated further, given their relevance both for mathematical and physical purposes.[2] Since gauge fields are typically sourced by defects, we then provide an alternative intuitive picture of how to encode the latter into a given bordism, by appropriately cutting out balls. We also point out the problem of how the cobordism conjecture should be capable of predicting Kaluza–Klein monopoles.

In Section 5, we consider the cobordism conjecture in the limit where gravity is decoupled. Taking this limit in a specific manner (maintaining compactness and sending the string coupling to zero) reveals a prominent role played by NSNS backgrounds, that we discuss further. Finally, in Section 6 we provide a summary and an outlook, with related remarks on T-duality as well as the finiteness conjecture.

# 2 Bordisms, cobordism conjecture and global symmetry: review

In this section, we first review the notions of bordism and bordism group. We then present the cobordism conjecture, and finally recall a few points on global symmetries to which the conjecture is related.

---

[2]While bordism groups for topological structures are typically discrete (such as $\mathbb{Z}$), one might expect that bordism groups for geometric structures are non-discrete (such as U(1)).

## 2.1 Bordisms and bordism groups

Two manifolds are bordant if they form the boundary of some other manifold. In this section we review how to make this precise, how equivalence classes of bordant manifolds give rise to bordism groups, and how additional topological structures can be incorporated into these constructions. For more background and details we refer to the textbook [23] and the lecture notes [24].

Let $M$ and $N$ be closed $k$-dimensional manifolds. A *bordism* between them is a compact $(k+1)$-dimensional manifold $W$ together with a diffeomorphism $\partial W \cong M \sqcup N$ between the boundary of $W$ and the disjoint union of $M$ and $N$. If there is such a bordism, then $M$ and $N$ are called *bordant*. This is an equivalence relation: reflexivity is witnessed by the cylinder $W = [0,1] \times M$, symmetry follows from the diffeomorphism $M \sqcup N \cong N \sqcup M$, and transitivity holds because if $\partial W \cong M \sqcup N$ and $\partial \widetilde{W} \cong L \sqcup M$, then the gluing of $W$ and $\widetilde{W}$ along their boundary components corresponding to $M$ is a bordism between $L$ and $N$.

The set of equivalence classes $[M]$ of $k$-dimensional bordant manifolds is denoted $\Omega_k^O$, or simply $\Omega_k$. It has the structure of an abelian group whose addition is induced by disjoint union. Indeed, if $W$ is a bordism between $M$ and $N$ while $W'$ is a bordism between $M'$ and $N'$, then $W \sqcup W'$ is a bordism between $M \sqcup M'$ and $N \sqcup N'$. Hence the addition

$$\Omega_k \times \Omega_k \longrightarrow \Omega_k, \quad ([M], [M']) \longmapsto [M \sqcup M'] \tag{2.1}$$

is well-defined, with respect to which the empty set $\varnothing$ (viewed as a $k$-dimensional manifold) represents the neutral element, $[\varnothing \sqcup M] = [M]$. By definition, the *(unoriented) bordism group (in dimension $k$)* is $\Omega_k$ together with the addition (2.1). Viewing the cylinder $[0,1] \times M$ as a bordism between $\partial([0,1] \times M) \cong M \sqcup M$ and $\varnothing$ shows that $[M]$ is its own inverse for all $[M] \in \Omega_k$; the following picture illustrates the identity $[M \sqcup M] = [\varnothing]$ in $\Omega_k$:

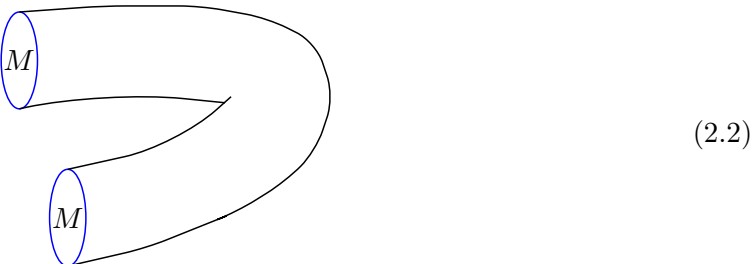

$$\tag{2.2}$$

As a first example, let us consider $k = 0$ and observe that a closed 0-dimensional manifold is a finite disjoint union of a single point pt with itself. Two such manifolds $M = \mathrm{pt}^{\sqcup m}$ and $N = \mathrm{pt}^{\sqcup n}$ with $m, n \geqslant 0$ are bordant iff $m + n$ is even, hence $\Omega_0 = \mathbb{Z}_2$. To see this, we note that the interval $[0,1]$ with $\partial[0,1] = \{0,1\} \cong \mathrm{pt} \sqcup \mathrm{pt}$ can be viewed both as a bordism between pt and pt, or between $\mathrm{pt} \sqcup \mathrm{pt}$ and $\varnothing$. In this way all but possibly one of the $m + n$ points in $\mathrm{pt}^{\sqcup m} \sqcup \mathrm{pt}^{\sqcup n}$ can be paired up:

$$\left[\mathrm{pt}^{\sqcup m} \sqcup \mathrm{pt}^{\sqcup n}\right] = \begin{cases} [\varnothing] & \text{if } m + n \text{ is even} \\ [\mathrm{pt}] & \text{otherwise.} \end{cases} \tag{2.3}$$

Similarly one finds that $\Omega_1 = 0$. Indeed, every 1-dimensional closed manifold is a finite

disjoint union of circles $S^1$, and $S^1$ bounds the disc $B_2$, hence $[S^1] = [\varnothing]$:

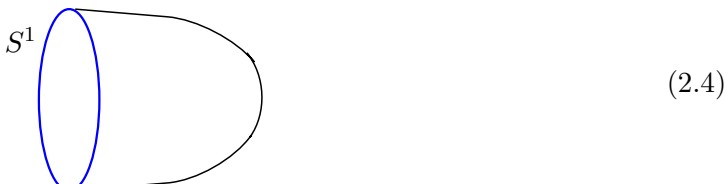

$$(2.4)$$

Another example is $\Omega_2 = \mathbb{Z}_2$. Proving this is standard but less elementary. First of all, the 2-sphere $S^2$ is the boundary of the 3-ball $B_3$, hence $[S^2] = [\varnothing]$ in $\Omega_2$. Similarly, by cutting a 2-torus $T^2$ out of $B_3$, we obtain a bordism between $S^2$ and $T^2$ and hence $[S^2] = [T^2]$:

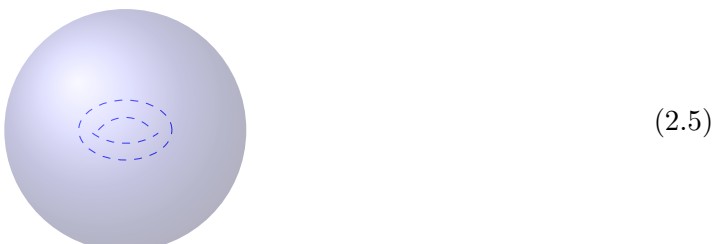

$$(2.5)$$

In general, every closed surface is a finite disjoint union of orientable surfaces and non-orientable surfaces. The former are either spheres or tori glued together along the boundaries of excised discs, i.e. connected sums $T^2 \# \ldots \# T^2$. Moreover, every connected closed non-orientable surface is a connected sum $\mathbb{RP}^2 \# \ldots \# \mathbb{RP}^2$ of projective planes, and $[\mathbb{RP}^2]$ turns out to be the non-trivial element in $\Omega_2$.

Elements in the bordism group $\Omega_k$ are represented by plain manifolds $M$. We are however often interested in additional structures such as principal $G$-bundles over $M$, for some Lie group $G$. To include such structures in our context, we first recall the definition of the refined bordism groups $\Omega_k(X)$ for any space $X$; then we describe how this in particular captures the case of $G$-bundles (by choosing $X$ to be the classifying space of $G$ as described below).

Let $X$ be a topological space, and let $M, N$ be $k$-dimensional closed manifolds. Two continuous maps $f\colon M \longrightarrow X$ and $g\colon N \longrightarrow X$ are *bordant* if there is a bordism $W$ between $M$ and $N$ together with a continuous map $F\colon W \longrightarrow X$ which restricts to $f$ and $g$ at the boundary. Again, being bordant is an equivalence relation, and we write $\Omega_k(X)$ for the set of equivalence classes $[M, f]$ of continuous maps $f\colon M \longrightarrow X$; taking disjoint union endows $\Omega_k(X)$ with the structure of an abelian group.

In the special case when $X = \mathrm{pt}$ is a single point, there is only a single map $M \longrightarrow \mathrm{pt}$ for each manifold $M$. Hence we recover the bordism group $\Omega_k = \Omega_k(\mathrm{pt})$. In general one finds that the operations $X \longmapsto \Omega_k(X)$ form a generalised homology theory with coefficients in the abelian groups $\Omega_k$.

To make the connection to principal $G$-bundles (which we usually simply refer to as $G$-bundles), recall that for every Lie group $G$, we have a contractible space $EG$ with a free action of $G$, a classifying space $BG$ (unique only up to homotopy), and a universal $G$-bundle $EG \longrightarrow BG$. The universality lies in the theorem that isomorphism classes of $G$-bundles are in bijection with homotopy classes of continuous maps $M \longrightarrow BG$. Concretely, every $G$-bundle $P \longrightarrow M$ is isomorphic to the pullback $c_P^*(EG)$ for some $c_P\colon M \longrightarrow BG$, which is called the *classifying map* of the bundle $P \longrightarrow M$. In particular, for every $k$-dimensional

manifold $M$ there is a classifying map

$$c_{TM}\colon M \longrightarrow B\mathrm{O}(k) \tag{2.6}$$

which corresponds to the tangent bundle $TM \longrightarrow M$, viewed as an $\mathrm{O}(k)$-bundle. In this case the classifying space $B\mathrm{O}(k)$ can be taken to be the space $\mathrm{Gr}_k(\mathbb{R}^\infty)$ of $k$-planes in $\mathbb{R}^\infty$, which can be thought of as the limit of the Grassmannian spaces $\mathrm{Gr}_k(\mathbb{R}^n)$ for $n \longrightarrow \infty$; precisely, $\mathrm{Gr}_k(\mathbb{R}^\infty) \simeq B\mathrm{O}(k)$ is defined as the "colimit" of the inclusions

$$\mathrm{Gr}_k(\mathbb{R}^k) \hookrightarrow \mathrm{Gr}_k(\mathbb{R}^{k+1}) \hookrightarrow \mathrm{Gr}_k(\mathbb{R}^{k+2}) \hookrightarrow \dots, \tag{2.7}$$

which to a good approximation we can think of as "$\lim_{q\to\infty} \mathrm{Gr}_k(\mathbb{R}^{k+q})$". The connection between $G$-bundles and the refined bordism group $\Omega_k(X)$ is now obtained by taking $X = BG$. We thus find that elements $[M, c]$ of the bordism group $\Omega_k(BG)$ are equivalently represented by manifolds $M$ with a $G$-bundle classified by $c\colon M \longrightarrow BG$, and $[M, c] = [M', c']$ iff there is a bordism with $G$-bundle structure that restricts to that of $M$ and $M'$ on the boundary.

In physics, we often need further structures on manifolds in addition to bundles. For example, fermions require spacetime to have a spin structure. This is an example of a "tangential structure" on a manifold, in the sense that it involves additional structure having to do with the tangent bundle (and the spin group, in the case of spin structures). The bordism groups $\Omega_k(X)$ can be further generalised by taking into account so-called "stable tangential structures", as we recall next. To the unfamiliar eye this discussion is rather technical, but we will see that it in particular allows us to consider oriented, spin or string manifolds on the same conceptual footing, and intuition for such more familiar structures can be carried over to the general case. Moreover, we will recall how the simplest tangential structure gives meaning to the superindex O in $\Omega_k = \Omega_k^{\mathrm{O}}$.

First, a *$k$-dimensional tangential structure* is a pointed topological space $\mathcal{X}(k)$ together with a pointed fibration $\xi_k\colon \mathcal{X}(k) \longrightarrow B\mathrm{O}(k)$. For example, given a Lie group $G$ and a continuous group homomorphism $\varphi\colon G \longrightarrow \mathrm{O}(k)$, we can set $\mathcal{X}(k) = BG$ and $\xi_k = B\varphi$ (using the fact that "taking classifying spaces" is functorial[3], which means that every continuous group homomorphisms $\psi\colon G \longrightarrow G'$ is mapped to a continuous map $B\psi\colon BG \longrightarrow BG'$ between classifying spaces, and that this mapping is compatible with composition). Then a *manifold with $\xi_k$-structure* (or: a *$\xi_k$-manifold*) is a $k$-dimensional manifold $M$ together with a lift of its classifying map $c_{TM}$ across $\xi_k$, i.e. a continuous map $c_{\xi_k}\colon M \longrightarrow \mathcal{X}(k)$ such that

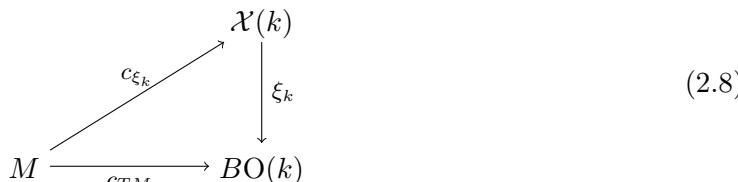

$$\tag{2.8}$$

commutes up to homotopy. Clearly, the relation to tangent bundles is provided by the discussion around (2.6) above.

---

[3]Indeed, there is a functor $B$ from topological groups to topological spaces given by the composition of delooping, taking the nerve, and taking geometric realisation. For the group $\mathrm{O}(k)$, this produces a model of the classifying space $B\mathrm{O}(k)$ different from (but homotopy equivalent to) the colimit of (2.7).

One finds that the special case of a $\xi_k$-structure on $M$ which is induced by a group homomorphism $G \longrightarrow \mathrm{O}(k)$ is equivalent to a choice of principal $G$-bundle $P \longrightarrow M$ together with an isomorphism of principal $\mathrm{O}(k)$-bundles between the associated bundle $P \times_G \mathrm{O}(k)$ and the bundle of orthonormal frames on $M$. Hence in particular a tangential structure coming from the inclusion $\mathrm{SO}(k) \hookrightarrow \mathrm{O}(k)$ is equivalent to an orientation on $M$, and a tangential structure coming from the double cover $\mathrm{Spin}(k) \longrightarrow \mathrm{SO}(k)$ post-composed with $\mathrm{SO}(k) \hookrightarrow \mathrm{O}(k)$ is equivalent to a spin structure on $M$. More generally, we refer to a $\xi_k$-structure that arises from a group homomorphism $\varphi \colon G \longrightarrow \mathrm{O}(k)$ as a *G-structure* (leaving $\varphi$ implicit). In physics, $G$-structures are particularly important, but for example in Section 3 below we will encounter other tangential structures that may also play an important role in quantum gravity.

In connection with bordism groups, of particular interest are so-called *stable tangential structures*, to which we will restrict the discussions in the present paper. Such structures are defined in terms of the stable orthogonal group

$$\mathrm{O} \equiv \mathrm{O}(\infty) := \mathrm{colim}\big( \mathrm{O}(1) \hookrightarrow \mathrm{O}(2) \hookrightarrow \mathrm{O}(3) \hookrightarrow \dots \big), \qquad (2.9)$$

namely as a pointed topological space $\mathcal{X}$ together with a pointed fibration $\xi \colon \mathcal{X} \longrightarrow B\mathrm{O}$. From this we obtain an associated $k$-dimensional tangential structure $\xi_k \colon \mathcal{X}(k) \longrightarrow B\mathrm{O}(k)$ as the pullback

$$\begin{array}{ccc} \mathcal{X}(k) & \longrightarrow & \mathcal{X} \\ {\scriptstyle \xi_k} \downarrow & & \downarrow {\scriptstyle \xi} \\ B\mathrm{O}(k) & \hookrightarrow & B\mathrm{O} \end{array} \qquad (2.10)$$

for any $k \geqslant 0$. Then by definition a *(stable) $\xi$-structure* on a $k$-dimensional manifold (or: a *$\xi$-manifold*) $M$ is a $\xi_k$-structure on $M$.

Orientation and spin are examples of stable tangential structures, namely $B\mathrm{SO} \longrightarrow B\mathrm{O}$ from the inclusion $\mathrm{SO} \hookrightarrow \mathrm{O}$, and $B\mathrm{Spin} \longrightarrow B\mathrm{O}$ from $\mathrm{Spin} \longrightarrow \mathrm{SO} \hookrightarrow \mathrm{O}$, where the stable groups $\mathrm{SO}$ and $\mathrm{Spin}$ are defined analogously to $\mathrm{O}$. Two further examples are the stable framing $\mathrm{pt} \simeq E\mathrm{O} \longrightarrow B\mathrm{O}$, and the stable tangential structure $\mathrm{id}_{B\mathrm{O}} \colon B\mathrm{O} \longrightarrow B\mathrm{O}$, or equivalently an $\mathrm{O}$-structure, which (up to homotopy) is really no structure at all.

The familiar notion of the "opposite of an orientation" has a natural generalisation to arbitrary stable tangential structures, which we will use below to identify inverses in generalised bordism groups. Here the basic idea is to use the reflection map $x \longmapsto -x$ for real numbers $x$, which in particular reverses a given orientation on $\mathbb{R}$. To discuss the opposites in general, note that if a $(k+1)$-dimensional manifold $W$ has a non-trivial boundary, there are two trivialisations of the normal bundle leading to $TW \cong \underline{\mathbb{R}} \oplus T(\partial W)$, and hence two ways for a $\xi$-structure on $W$ to induce a $\xi$-structure on $\partial W$. We use the convention that *the $\xi$-structure on $\partial W$* is the one obtained using the outward normal, while we denote the boundary endowed with the other $\xi$-structure as $-\partial W$. For example, if $\xi$ comes from the inclusion $\mathrm{SO} \hookrightarrow \mathrm{O}$, then $-\partial W$ has the opposite orientation.

Having recalled the definition of (opposite) tangential structure, we are now prepared to consider further refinements of bordism groups. First we will introduce the refined version for general tangential structures, then we discuss the case of $G$-structures in more detail. Let $\xi \colon \mathcal{X} \longrightarrow B\mathrm{O}$ be a stable tangential structure, and let $M, N$ be $k$-dimensional $\xi$-manifolds.

| $k$ | 0 | 1 | 2 | 3 | 4 | 5 | 6 | 7 | 8 | 9 | 10 |
|---|---|---|---|---|---|---|---|---|---|---|---|
| $\Omega_k^{\mathrm{fr}}$ | $\mathbb{Z}$ | $\mathbb{Z}_2$ | $\mathbb{Z}_2$ | $\mathbb{Z}_{24}$ | $0$ | $0$ | $\mathbb{Z}_2$ | $\mathbb{Z}_{240}$ | $\mathbb{Z}_2^2$ | $\mathbb{Z}_2^3$ | $\mathbb{Z}_6$ |
| $\Omega_k^{\mathrm{O}}$ | $\mathbb{Z}_2$ | $0$ | $\mathbb{Z}_2$ | $0$ | $\mathbb{Z}_2^2$ | $\mathbb{Z}_2$ | $\mathbb{Z}_2^3$ | $\mathbb{Z}_2$ | $\mathbb{Z}_2^5$ | $\mathbb{Z}_2^3$ | $\mathbb{Z}_2^8$ |
| $\Omega_k^{\mathrm{SO}}$ | $\mathbb{Z}$ | $0$ | $0$ | $0$ | $\mathbb{Z}$ | $\mathbb{Z}_2$ | $0$ | $0$ | $\mathbb{Z}^2$ | $\mathbb{Z}_2^2$ | $\mathbb{Z}_2$ |
| $\Omega_k^{\mathrm{Spin}}$ | $\mathbb{Z}$ | $\mathbb{Z}_2$ | $\mathbb{Z}_2$ | $0$ | $\mathbb{Z}$ | $0$ | $0$ | $0$ | $\mathbb{Z}^2$ | $\mathbb{Z}_2^2$ | $\mathbb{Z}_2^3$ |
| $\Omega_k^{\mathrm{Spin}^c}$ | $\mathbb{Z}$ | $0$ | $\mathbb{Z}$ | $0$ | $\mathbb{Z}^2$ | $0$ | $\mathbb{Z}^2$ | $0$ | $\mathbb{Z}^4$ | $0$ | $\mathbb{Z}_2 \times \mathbb{Z}^4$ |
| $\Omega_k^{\mathrm{Pin}^+}$ | $\mathbb{Z}_2$ | $0$ | $\mathbb{Z}_2$ | $\mathbb{Z}_2$ | $\mathbb{Z}_{16}$ | $0$ | $0$ | $0$ | $\mathbb{Z}_2 \times \mathbb{Z}_{32}$ | $0$ | $\mathbb{Z}_2^3$ |
| $\Omega_k^{\mathrm{String}}$ | $\mathbb{Z}$ | $\mathbb{Z}_2$ | $\mathbb{Z}_2$ | $\mathbb{Z}_{24}$ | $0$ | $0$ | $\mathbb{Z}_2$ | $0$ | $\mathbb{Z}_2 \times \mathbb{Z}$ | $\mathbb{Z}_2^2$ | $\mathbb{Z}_6$ |

Table 2.1: Bordism groups $\Omega_k^\xi$ for various dimensions $k$ and $\xi$-structures.

A $\xi$-*bordism* from $M$ to $N$ is a $(k+1)$-dimensional compact $\xi$-manifold $W$ together with a decomposition $\partial W \cong W_1 \sqcup W_2$ and diffeomorphisms of $\xi$-manifolds $M \cong -W_1$ and $N \cong W_2$. For example the cylinder $W = [0,1] \times M$ has a natural $\xi$-structure such that $W_1 := \{0\} \times M \cong -M$ and $W_2 := \{1\} \times M \cong M$, making $W$ into a bordism from $M$ to itself. Alternatively, we can view the same $\xi$-manifold $[0,1] \times M$ as a bordism from $(-M) \sqcup M$ to the empty set, by setting $W_1 := M \sqcup (-M)$ and $W_2 := \varnothing$. Here for a closed $\xi$-manifold $M$, we denote by $-M$ the same underlying manifold with the opposite $\xi$-structure, namely the one induced on $\{0\} \times M$ by the bordism $[0,1] \times M$ with $\xi$-structure such that $\{1\} \times M$ corresponds to the $\xi$-manifold $M$. For $k = 1$, $M = S^1$ and $\xi$ corresponding to orientation, the two interpretations of the cylinder $[0,1] \times M$ can be illustrated as follows:

$$: M \longrightarrow M, \tag{2.11}$$

$$: M \sqcup (-M) \longrightarrow \varnothing. \tag{2.12}$$

Admitting a $\xi$-bordism is again an equivalence relation between $k$-dimensional closed $\xi$-manifolds, and again disjoint union provides the set of equivalence classes with the structure of an abelian group. We denote this $\xi$-*bordism group* by $\Omega_k^\xi$, or as $\Omega_k^G$ if $\xi$ comes from a group homomorphism $G \longrightarrow \mathrm{O}$, or as $\Omega_k^{\mathrm{fr}}$ if $\xi$ is the fibration $E\mathrm{O} \longrightarrow B\mathrm{O}$. Again $[\varnothing]$ is the neutral element of $\Omega_k^\xi$, and the discussion of the previous paragraph shows that $[-M]$ is inverse to $[M]$ in $\Omega_k^\xi$, because $[(-M) \sqcup M] = [\varnothing]$.

In Table 2.1 we collect a number of examples of $\xi$-bordism groups, reproduced from [25,26] and the references given in [4, App. A]. For instance we have $\Omega_2^{\mathrm{SO}} = 0$, as we already saw in

our discussion of the unoriented bordism group $\Omega_2 = \mathbb{Z}_2$, cf. the text around (2.5). Another simple example is $\Omega_0^{\mathrm{SO}} = \mathbb{Z}$, which is generated by the class of the positively oriented point $+$ (whose inverse is the class of the negatively oriented point $-$). To see this it is crucial that the diffeomorphisms which are part of a $\xi$-bordism must be compatible with $\xi$; in the case of $B\mathrm{SO} \longrightarrow B\mathrm{O}$ this means orientation-preserving, so in particular there is no oriented bordism between $+$ (corresponding to $+1 \in \mathbb{Z}$) and $-$ (corresponding to $-1 \in \mathbb{Z}$).

Given a stable tangential structure $\xi\colon \mathcal{X} \longrightarrow B\mathrm{O}$ and a topological space $Y$, one can straightforwardly combine the constructions of $\Omega_k^\xi$ and $\Omega_k(Y)$ into the definition of a bordism group $\Omega_k^\xi(Y)$. This is however no real synthesis, as one can show that there is a canonical tangential structure $\xi_Y$ associated to $\xi$ and $Y$ such that $\Omega_k^\xi(Y) \cong \Omega_k^{\xi_Y}$. Hence every bordism group we have discussed so far is of the form $\Omega_k^\xi$ for some tangential structure $\xi$.[4]

As a side remark, let us mention that bordism groups $\Omega_k^\xi$ are part of the richer structure of bordism categories $\mathrm{Bord}_{k+1,k}^\xi$. The latter have closed $k$-dimensional $\xi$-manifolds as objects and diffeomorphism classes of $\xi$-bordisms between them as morphisms. Hence two manifolds represent the same element in $\Omega_k^\xi$ if and only if there is a morphism between them in $\mathrm{Bord}_{k+1,k}^\xi$. Put differently, bordism groups are the connected components of bordism categories,

$$\Omega_k^\xi = \pi_0\big(\mathrm{Bord}_{k+1,k}^\xi\big). \tag{2.13}$$

Bordism categories are for example relevant in the functorial description of topological quantum field theories, which are certain functors on $\mathrm{Bord}_{k+1,k}^\xi$.

We are now prepared to discuss the cobordism conjecture, which is a statement that relates the geometric structures relevant in quantum gravity with bordism groups.

## 2.2 The cobordism conjecture

As part of the swampland program, the framework for the conjecture is the study of effective theories of quantum gravity in a $D$-dimensional extended spacetime. More precisely, we will focus on theories obtained by a compactification (for example of string theory) on a $k$-dimensional closed manifold $M$, i.e. $M$ is compact and $\partial M = \varnothing$. (In this paper, all manifolds we consider are assumed to be smooth.) In such a compactification from $D + k$ to $D$ dimensions, the relevant information is the topology and geometry of $M$, and potentially further data that we denote $\mathcal{D}$ for convenience. These extra data can for instance be a tangential $\xi$-structure, or possible fluxes on $M$ (which we discuss further in Section 4.2 below). As explained in Section 1, one motivation behind the cobordism conjecture [4] is to claim that all compactifications on $(M, \mathcal{D})$ and the resulting $D$-dimensional theories are related to each other, and this relation is established by the existence of certain bordisms. More precisely, the cobordism conjecture postulates the existence of a "quantum gravity structure", denoted QG, and states that the associated bordism group is trivial:

$$\Omega_k^{\mathrm{QG}} = 0. \tag{2.14}$$

---

[4]For completeness, we should also mention the Pontryagin–Thom construction, which for every stable tangential structure $\xi\colon \mathcal{X} \longrightarrow B\mathrm{O}$ produces isomorphisms $\Omega_k^\xi \cong \pi_k(M\mathcal{X})$, where the right-hand side is the $k$-th stable homotopy group of the Thom spectrum $M\mathcal{X}$ associated to $\xi$. In the special case of stable framings $E\mathrm{O} \longrightarrow B\mathrm{O}$, the Thom spectrum is equivalent to the sphere spectrum $\mathbb{S} = M\mathbb{1}$, whose $n$-th space is the $n$-sphere $S^n$. Hence the bordism groups $\Omega_k^{\mathrm{fr}}$ are isomorphic to the $k$-th stable homotopy groups of spheres, $\Omega_k^{\mathrm{fr}} \cong \pi_k(\mathbb{S}) \cong \pi_{k+n}(S^n)$ for all $n > k + 1$.

This triviality implies that there exists a single QG-bordism equivalence class, and therefore that all of its representatives $(M, \mathcal{D})$ are bordant to each other. In this sense, all compactifications are thus related. Note that the bordism can be along the time direction, in which case it can be understood as a dynamical process, but this is not necessarily the case. For instance dualities are typically not viewed as dynamical processes, and may rather be implemented by spatial bordisms.

The difficulty of the conjecture is that the structure QG is not known explicitly. To verify whether the statement (2.14) can hold, what is proposed in [4] is to first study examples, and identify which $\xi$-structures allow for a trivialisation of the corresponding bordism group. For instance, Calabi–Yau 3-folds are known to be valid 6-dimensional closed manifolds for a string compactification. Those admit a spin structure, and one has $\Omega_6^{\text{Spin}} = 0$. The same goes for $G_2$-manifolds in M-theory compactifications, and one as $\Omega_7^{\text{Spin}} = 0$. Therefore, a spin structure might naively be considered as candidate for QG in (2.14). However, as can be seen in Table 2.1, $\Omega_k^{\text{Spin}} \neq 0$ for some $k$. In the cases where the bordism group does not vanish, the idea is to look for another $\xi$-structure that will reduce the order of the bordism group, or even make it trivial; in this last case, the initial non-trivial bordism group is said to be "killed". Adapting the $\xi$-structure in this manner, one hopes to get closer to QG. Reducing the order of the group means collecting the allowed $(M, \mathcal{D})$ in fewer classes, therefore connecting more compactifications via a bordism, in line with the motivation behind the cobordism conjecture.

Let us give a few examples of this procedure from [4]. Considering a circle, one has for instance $\Omega_1^{\text{Spin}} = \mathbb{Z}_2$, which can be killed through $\Omega_1^{\text{Pin}^+} = 0$: this change of structure occurs when allowing compactification of type IIA string theory on unoriented manifolds, or equivalently when including orientifold $O_8$-planes. Similarly, one finds that $[K3]$ is a generator of $\Omega_4^{\text{Spin}} = \mathbb{Z}$. This group can be reduced to $\Omega_4^{\text{Pin}^+} = \mathbb{Z}_{16}$ by compactifying M-theory on unoriented manifolds, and it is further decreased by including M-theory orientifolds, $MO_5$-planes. This inclusion of $O_p$-planes or more generally extended objects, or defects, is a point we will come back to in Section 4.

To summarise, in order to test (2.14) but also to determine QG, the procedure consists in changing the topological or geometric structure, whenever first facing a non-trivial bordism group in a valid compactification, i.e. a potential counterexample to the conjecture. Such a modification may amount to a new *topological* structure like the tangential $\xi$-structures discussed in Section 2.1; but it may also lead to *geometric* structures like Riemannian metrics or gauge connections, which we will discuss in Section 4.2. Moreover, any change of the bordism group should however admit a physical interpretation, such as allowing for different manifolds or including defects with a concrete physical description. Interestingly, when no obvious physical interpretation is found for the change, enforcing a trivial bordism group can result in the prediction of new quantum gravity objects or properties [4,14].

Another important aspect of the cobordism conjecture (2.14), independent of the definition of QG, is that the single equivalence class is the neutral element, whose representative we can denote as $(\varnothing, 0)$. The physical interpretation is that any compactification is connected to the empty set without any physical content (no flux or charge, etc., hence $\mathcal{D} = 0$). Since the empty set has no point, it is very reminiscent of the physical concept of bubbles of nothing [17,18,27–29]. Those can be understood as voids "appearing" in space, and sometimes "growing" in the sense that the volume of the initial space diminishes, and space ends up completely disappearing. The cobordism conjecture can be interpreted as saying that any compactification is related to *nothing*, i.e. the final state of a growing bubble of nothing.

This may appear physically as dramatic, but one should note that there is no information on the dynamics of such a process. In particular, bubbles of nothing may need a certain minimal energy to be activated, or excited as a state, which could be dynamically prohibited by the theory. Information about dynamics of quantum gravity is absent in the cobordism conjecture, and should rather be searched for in other swampland conjectures [17], or elsewhere. From this perspective, the cobordism conjecture "only" allows us to characterise all possible compactifications and quantum gravity ingredients, without each of them being necessarily reached. Still, such a characterisation can provide interesting constraints on compactification spaces. For example, combining the cobordism conjecture with other swampland considerations, in [7, 14, 30] it is shown that not only the rank, but the actual gauge algebra of theories with 16 supercharges in $d > 6$ dimensions is highly constrained, and most, if not all, of those supergravity constructions which are not realised in string theory are in the swampland.

Last but not least, an important motivation for the cobordism conjecture comes from another swampland conjecture: the absence of global symmetry in quantum gravity. We now turn to the relation between the two, emphasising the role of $(\varnothing, 0)$.

## 2.3   No global symmetry

### 2.3.1   Relation to the cobordism conjecture

One expectation of a quantum gravity theory is the absence of global symmetries. This is an old idea (see [31] and references therein, in particular [32, 33]) that has been revisited more rigorously recently in [34–36] with interesting new input e. g. in [37–41]. This statement does not rule out accidental or emergent global symmetries in low-energy effective field theories; but those should not be present any more, for instance by being broken or gauged, when reaching a UV completion within quantum gravity. The original argument against a global symmetry, that we briefly recall below, has to do with black hole evaporation, hence the connection to quantum gravity. The cobordism conjecture can be viewed as inspired by this first conjecture, and in turn implies it, as we will explain in the following as well as in Section 4.

In short, the standard argument with black holes goes as follows. Consider a physical state charged under a global symmetry, and let it fall into a black hole; the black hole then carries this global charge. The latter cannot be annihilated through Hawking radiation at the horizon (see e. g. [42]), so it remains while the black hole evaporates. At the end of this process, because of the global charge, the black hole cannot completely disappear, otherwise leading to information loss. This implies that it should eventually turn into a remnant carrying the global charge. Black hole remnants are however typically undesired in physics, one reason being that they have not been observed in nature. Assuming their absence, one concludes on the absence of a global charge, thus of a global symmetry.

That argument can be related to the cobordism conjecture as follows. Let us consider the ball made of the interior of the black hole and delimited by a spherical horizon: this compact space is the only part of space that matters in the argument. The complete evaporation of the black hole without any final remnant physically means that the black hole disappears, and the ball just defined shrinks to nothing. This dynamical process can be described as a bordism (along time) between the initial compact space (the ball and the spherical horizon) and the empty set (nothing). An illustration is the figure in (2.4) where the circle represents the spherical horizon. In turn, imposing as in the cobordism conjecture (2.14) the existence of a bordism between the initial compact space and the empty set physically implies the absence

of a remnant, thus the impossibility of a global charge and the absence of a global symmetry. In that sense, the cobordism conjecture disallows global symmetries. We will come back to this point in Section 4.

Studying the cobordism conjecture, one is lead to consider various numbers $n_M$ on compact manifolds $M$, such as values of characteristic classes (see Section 3) or flux numbers (see Section 4): let us call those collectively "topological numbers". Such a number $n_M$ is analogous to a global charge carried by a compact manifold: this analogy allows to mimic the previous black hole argument. Imposing that $M$ is bordant to the empty set can be problematic because the latter cannot carry any non-zero topological number. Such a situation has two possible outcomes. The first one is to say that the empty set requires to have $n_M = 0$: this argument is analogous to the absence of a remnant, implying the absence of a global charge. The second one is to require the presence of something else, e.g. a localised compensating physical object carrying the same $n_M$, i.e. a charged defect. This would be analogous to a black hole remnant with a global charge, if one would allow for their existence. In the following sections, we will encounter both outcomes. Interestingly, the existence of most $\xi$-structures mentioned above require the vanishing of a characteristic class, and a corresponding vanishing topological number. For instance, orientation corresponds to a vanishing first Stiefel–Whitney class $w_1(M) = 0$, a spin structure requires in addition $w_2(M) = 0$, and a string structure also asks for a vanishing fractional first Pontryagin class, $\frac{1}{2}p_1(M) = 0$. As will be discussed further in Section 3, those vanishing numbers have in addition interesting physical interpretations in terms of anomalies in string theory. Finding the appropriate QG structure seems therefore to cancel the relevant topological numbers, thus going again, at least by analogy, in the direction of forbidding global charges and global symmetries.

### 2.3.2 Gauging a global symmetry

We finally provide some further comments on global symmetries, and how to gauge them, restricting to U(1) for simplicity. In particular, we review here how to interpret $dF_q = 0$, for a $q$-form field strength $F_q$, as the consequence of a global symmetry [39]. On general grounds, one can associate a conserved Noether current to any continuous global symmetry. In particular, a generalised $(d - k - 1)$-form global symmetry (see [43]) admits a conserved $k$-form Noether current $j_k$, such that $dj_k = 0$. Therefore, the $k$-form $F_k$ such that $dF_k = 0$ can be interpreted as a conserved current associated to a $(d - k - 1)$-form global symmetry. A minimal model realising this would be

$$S \sim \int -\frac{1}{2} F_k \wedge *F_k \,, \qquad *F_k = d\tilde{A}_{d-k-1} \,, \tag{2.15}$$

where we choose the fundamental field to be $\tilde{A}_{d-k-1}$. Varying the action $S$ with respect to $\tilde{A}$, we get

$$dF_k = 0 \,, \tag{2.16}$$

i.e. a conserved current $j_k = F_k$ associated to the $(d - k - 1)$-form global symmetry $\tilde{A}_{d-k-1}$.

We now turn to the gauging. By introducing an additional source term in the action,

$$S \sim \int \left( -\frac{1}{2} F_k \wedge *F_k + j_{k+1} \wedge \tilde{A}_{d-k-1} \right) \,, \tag{2.17}$$

we gauge the $(d - k - 1)$-form global symmetry. Indeed, the action is now invariant under a local transformation $\tilde{A}_{d-k-1} \longmapsto \tilde{A}_{d-k-1} + d\Lambda_{d-k-2}$, given that $dj_{k+1} = 0$. This holds since

$j_{k+1}$ is exact, as variation with respect to $\tilde{A}$ yields

$$j_{k+1} = (-1)^{k+1} \, \mathrm{d}F_k \,. \tag{2.18}$$

In turn, since $j_{k+1} \neq 0$, one deduces $\mathrm{d}F_k \neq 0$ so the original global symmetry is not present any more. Note that we could have had an equivalent discussion starting from the fundamental field $A_{k-1}$, with electromagnetic duality $A \longleftrightarrow \tilde{A}$.

The above discussion will be useful in Section 4 to relate once again the cobordism conjecture to the absence of global symmetry. Before doing so, we present in the next section an interesting organisational principle of structures that may appear on the way to QG.

## 3   Whitehead tower for the orthogonal group

The cobordism conjecture states that there is a "quantum gravity structure" QG such that $\Omega_k^{\mathrm{QG}} = 0$ for all compact dimensions $k$, but it does not tell us what QG is. In this section, we begin to explore one particular approach that aims to provide ways to systematically approximate the unidentified structure QG. The main notion in the approach is the "White-head tower" of a given space, which constructs a series of spaces with more and more trivial homotopy groups. Simultaneously, it keeps track of possible obstructions for these spaces to be tangential structures for a given manifold. The main example we consider is the White-head tower of the orthogonal group O (which coherently organises orientations, spin, string and fivebrane structures, among other tangential structures), whose application to anomaly cancellation in string theory was pioneered in [44]. We will show that all bordism groups for this example are either of finite order or even trivial when climbing the Whitehead tower far enough, namely to "fivebrane level" or higher.

### 3.1   Mathematical background

We begin by summarising the construction for an arbitrary pointed CW complex $X$; for further details we refer e.g. to [45, Ch. 18]. Later, we will discuss the example $X = BO$ more explicitly. One advantage of leading with the abstract description is that it clarifies how obstructions are encoded into Whitehead towers.

Recall that for a non-negative integer $i$, the $i$-th homotopy group $\pi_i(X)$ by definition has homotopy classes of pointed continuous maps $S^i \longrightarrow X$ as elements, where the homotopies must be constant on the chosen basepoint of $S^i$. A space whose $n$-th homotopy group is a given abelian group $A$, while all other homotopy groups are trivial, is called an Eilenberg–Mac Lane space and denoted $K(A, n)$. Hence, by definition

$$\pi_n\big(K(A,n)\big) \cong A \,, \quad \pi_i\big(K(A,n)\big) = 0 \text{ for } i \neq n \,. \tag{3.1}$$

For given $n$ and $A$, Eilenberg–Mac Lane spaces are unique up to homotopy equivalence; one model is the $n$-fold delooping,

$$K(A,n) \simeq B^n A \,. \tag{3.2}$$

For example, we have $B\mathbb{Z} \simeq K(\mathbb{Z},1) \simeq \mathrm{U}(1)$ (since the fundamental group $\pi_1(S^1) \cong \mathbb{Z}$ is the only non-trivial homotopy group of the circle $S^1 \simeq \mathrm{U}(1)$), and hence $K(\mathbb{Z},n) \simeq B^{n-1}\mathrm{U}(1)$ for all $n \geqslant 1$.

An important property of Eilenberg–Mac Lane spaces is that they form a spectrum representing singular cohomology. This means that for every pointed CW complex $X$, there are natural isomorphisms

$$\big[ X, K(A,n) \big] \cong H^n(X; A) \tag{3.3}$$

between homotopy classes of maps into $K(A,n)$ and the $n$-th cohomology with coefficients in $A$.

The main idea behind Whitehead towers is that for every space $X$ we can construct a series of spaces $X_n$ whose higher homotopy groups agree with those of $X$, but whose lower homotopy groups are all trivial (or "killed off"). In particular, by construction we will find that if for some $n$ the $n$-th homotopy group of $X$ is already trivial, $\pi_n(X) = 0$, then the spaces $X_n$ and $X_{n-1}$ can be taken to be equal, $X_n = X_{n-1}$.

For the precise definition, we assume that $X$ is path-connected, i.e. $\pi_0(X) = 0$. Then, the *Whitehead tower of $X$* consists of topological spaces $X_n$ such that

$$\pi_i(X_n) \cong \begin{cases} 0 & \text{if } i \leqslant n \\ \pi_i(X) & \text{if } i > n \end{cases} \tag{3.4}$$

together with fibrations $X_n \longrightarrow X_{n-1}$ for all positive integers $n$ whose fibres are the Eilenberg–Mac Lane spaces $K(\pi_n(X), n-1)$,

$$K\big(\pi_n(X), n-1\big) \longrightarrow X_n \longrightarrow X_{n-1} \tag{3.5}$$

where $X_0 := X$. It follows that $X_n$ is an $n$-connected cover of $X$. Concretely, these fibrations can be obtained inductively by first attaching higher cells to $X_{n-1}$ in such a way that all homotopy groups above $n$ vanish, thus constructing a model for $K(\pi_n(X), n)$ from $X_{n-1}$. In the second step, $X_n$ can be defined as the space of paths from a fixed base point in $K(\pi_n(X), n)$ whose endpoints are in $X_{n-1} \subset K(\pi_n(X), n)$, leading to (3.5). From this we also obtain a homotopy fibration

$$X_n \longrightarrow X_{n-1} \longrightarrow K\big(\pi_n(X), n\big) \tag{3.6}$$

via delooping. Another way of putting this is that post-composing a map $f \colon M \longrightarrow X_{n-1}$ with $X_{n-1} \longrightarrow K(\pi_n(X), n)$ gives a homotopically trivial map, i.e. one homotopic to a constant map, iff $f$ factors through the fibration $X_n \longrightarrow X_{n-1}$ up to homotopy, i.e. there is a lift $\widetilde{f} \colon M \longrightarrow X_n$ such that:

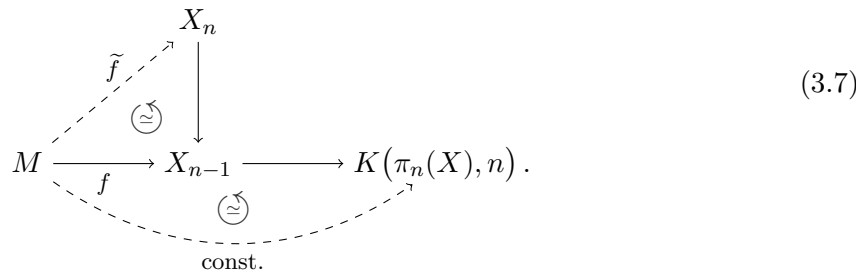

$$\tag{3.7}$$

In Section 3.2 we will see how this property encodes obstructions to the existence of tangential structures related to $X_n$, and discuss its physical and string theoretic interpretations.

We now focus on the case $X = BO$, the classifying space of the stable orthogonal group (2.9). By Bott periodicity, the homotopy groups of the latter are

$$\pi_i(\mathrm{O}) \cong \begin{cases} \mathbb{Z}_2 & \text{if } i \in \{0, 1\} \\ 0 & \text{if } i \in \{2, 4, 5, 6\} \\ \mathbb{Z} & \text{if } i \in \{3, 7\} \\ \pi_{i+8n}(\mathrm{O}) & \text{for all non-negative integers } n \end{cases} \tag{3.8}$$

and the homotopy groups of $\mathrm{O}(n)$ stabilise to those of $\mathrm{O}$ in the sense that $\pi_i(\mathrm{O}(n)) \cong \pi_i(\mathrm{O})$ for all $n > i + 1$. Moreover, delooping shifts homotopy groups in the sense that

$$\pi_{i+1}(BO) \cong \pi_i(\mathrm{O}). \tag{3.9}$$

The first twelve floors of the Whitehead tower of $BO$ are displayed in Figure 3.1, where we use the standard notation

$$BO\langle n \rangle := (BO)\langle n \rangle := (BO)_{n-1}, \tag{3.10}$$

where the index $n-1$ in the last term indicates the level in the Whitehead tower as introduced above. It follows that $BO\langle n \rangle \simeq B(\mathrm{O}\langle n - 1 \rangle)$ and

$$\pi_i(BO\langle n \rangle) \cong \begin{cases} 0 & \text{if } i < n \\ \pi_i(BO) \cong \pi_{i-1}(\mathrm{O}) & \text{if } i \geqslant n. \end{cases} \tag{3.11}$$

The homotopy groups $\pi_n(BO)$ appearing in the fibrations (3.5) and (3.6) in Figure 3.1 are obtained from (3.8) and (3.9), and we used (3.2) together with $K(\mathbb{Z}, 1) \simeq \mathrm{U}(1)$ and $K(\mathbb{Z}_2, 1) \simeq \mathbb{RP}^\infty$.

## 3.2 Physical interpretation

To discuss the labelled horizontal arrows in Figure 3.1, let us first consider the one labelled $w_1$, which is obtained by delooping the first fibration $BO\langle 2 \rangle = (BO)_1 = BSO \longrightarrow BO$. The associated diagram (3.7) for the classifying map $c_{TM} \colon M \longrightarrow BO$ of some manifold $M$ reads

$$\begin{array}{ccc} & BSO & \\ & \nearrow \Big\downarrow & \\ M \xrightarrow{\;c_{TM}\;} & BO \xrightarrow{\;w_1\;} & K(\mathbb{Z}_2, 1) \,. \end{array} \tag{3.12}$$

Hence $c_{TM}$ lifts to $BSO$, i.e. $M$ is orientable, iff $w_1 \circ c_{TM}$ is homotopically trivial. But according to (3.3) we have $[M, K(\mathbb{Z}_2, 1)] \cong H^1(M; \mathbb{Z}_2)$, so $M$ is orientable iff the associated cohomology class is trivial. This class is the first Stiefel–Whitney class $w_1(TM) \in H^1(M; \mathbb{Z}_2)$, which is the obstruction to orientability: $M$ is orientable iff $w_1(TM) = 0$.

Climbing the next few floors of the Whitehead tower, one finds that the second Stiefel–Whitney class $w_2(TM)$ is the obstruction for an oriented manifold $M$ to admit a spin structure, while the fractional first Pontryagin class $\frac{1}{2}p_1(TM)$ is the obstruction to lift a spin

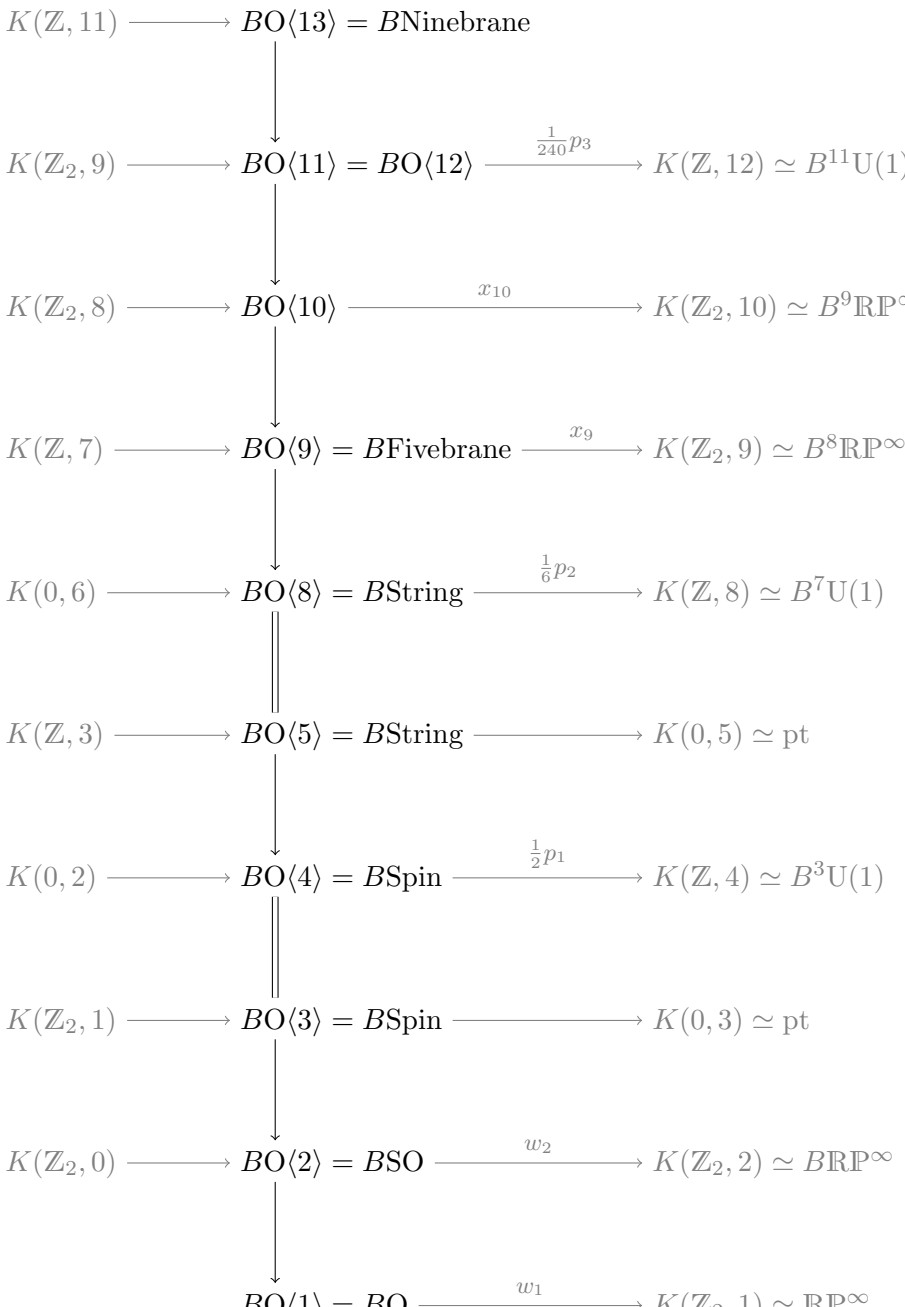

Figure 3.1: The first twelve floors of the Whitehead tower of $BO$, including the Eilenberg–Mac Lane spaces $K(\pi_{n-1}(BO), n-2)$ as the fibres of the fibrations $BO\langle n+1 \rangle \longrightarrow BO\langle n \rangle$ on the left, and the maps $BO\langle n \rangle \longrightarrow K(\pi_n(BO), n)$ encoding the obstructions to lifting $O\langle n-1 \rangle$-structures to $O\langle n \rangle$-structures on the right.

structure on $M$ to a string structure. The conditions $w_2(TM) = 0$ or $\frac{1}{2}p_1(TM) = 0$ can be interpreted as anomaly cancellation conditions on $M$ viewed as spacetime for a theory

of a spinning particle or type II string theory without gauge fields, respectively. This is reviewed in [44], where in addition $\frac{1}{2}p_2(TM)$ is identified as the obstruction to lifting a string structure on $M$ to an $O\langle 8\rangle$-structure, which is then interpreted within a generalised Green–Schwarz anomaly cancellation mechanism for a theory of fivebranes. To it, a string is an electric source to a $B$-field, giving a 3-form $H_3$-field, and the anomaly for the string structure appears as $dH_3 = \frac{1}{2}p_1$ in the absence of other gauge fields. Similarly, fivebranes are electric sources to a 7-form $H_7$-field, and the anomaly on the 6-dimensional brane worldvolume appears in $dH_7 = \frac{1}{6}p_2$, cf. [44]. Finally, the obstruction to lifting an $O\langle 10\rangle$-structure to an $O\langle 12\rangle$-structure is identified in [46] as $\frac{1}{240}p_3(TM)$ and related to anomaly cancellation for ninebranes. Hence $O\langle 8\rangle$- and $O\langle 12\rangle$-structures are referred to as fivebrane and ninebrane structures, respectively. A physical interpretation for $O\langle 9\rangle$- and $O\langle 10\rangle$-structures, respectively called 2-orientation and 2-spin in [46], has not yet been fully settled; we discuss a related proposal in Appendix A.

The relation between anomaly cancellation and obstructions to tangential structures that are encoded in the Whitehead tower of $BO$, as well as the neat conceptual organising principle that the tower offers, might induce hope that somewhere up in this tower we might find the "quantum gravity structure" QG. In the case of $BO$, we will now see that this idea leads to interesting features, yet ultimately it is too optimistic. This is because we should then determine the bordism groups related to the tangential $O\langle n+1\rangle$-structures associated to the $n$-th space in the Whitehead tower of $BO$, but as we will show momentarily we have

$$\Omega_k^{O\langle n+1\rangle} \cong \Omega_k^{\mathrm{fr}} \quad \text{for all } k \leqslant n. \tag{3.13}$$

A look at Table 2.1 indicates that most framed bordism groups for small values of $k$ are non-trivial, hence by $BO\langle n\rangle \simeq B(O\langle n-1\rangle)$ and (3.13), none of the components of the Whitehead tower for $BO$ can be the thought-after structure QG. This is made more explicit in Table 3.1 where we collect several bordism groups associated to the Whitehead tower of $BO$, in part obtained from the relation (3.13). Still, it is worth noting that $\Omega_k^{\mathrm{fr}}$ is only a finite group for all $k > 0$ (see e.g. [26, Lem. 1.1.8]). Thanks to (3.13), we are then guaranteed to reach more groups of finite order when going up the tower, which are at least closer to trivial than groups of infinite order such as $\Omega_4^{\mathrm{Spin}} = \mathbb{Z}$ or $\Omega_8^{\mathrm{String}} = \mathbb{Z}_2 \times \mathbb{Z}$.

To prove the existence of the isomorphisms (3.13), we first note that by construction $\pi_i(B(O\langle n+1\rangle)) \cong \pi_i(BO\langle n+2\rangle)$ for all $i \leqslant n+1$. Hence by the CW approximation theorem, the classification space $B(O\langle n+1\rangle)$ is homotopy equivalent to a CW complex that does not have any non-trivial cells of dimension $n+1$ or lower. This in turn implies that any continuous map $W \longrightarrow B(O\langle n+1\rangle)$ classifying an $O\langle n+1\rangle$-structure on a $(k+1)$-dimensional bordism $W$ induces trivial maps $\pi_i(W) \longrightarrow \pi_i(B(O\langle n+1\rangle))$ for all $i \leqslant n+1$. But since the $(k+1)$-dimensional manifold $W$ has no cells of dimension $k+2$ or larger, our assumption $k \leqslant n$ implies $\pi_i(W) = 0$ for all $i > n+1$. Thus every map $W \longrightarrow B(O\langle n+1\rangle)$ is homotopically trivial, so an $O\langle n+1\rangle$-structure on $W$ is equivalent to a tangential structure corresponding to the embedding of the trivial group into $O$, which in turn corresponds to a stable framing. This establishes (3.13).

While the Whitehead tower for $BO$ does not directly produce the unknown structure QG, it may still play a useful role in the search for it: instead of asking for all obstructions for a manifold to admit an $O\langle n\rangle$-structure to vanish, one may ask this of only some of the obstructions. For example, the conditions for a spin structure to exist on a manifold $M$ are

| $k$ | 0 | 1 | 2 | 3 | 4 | 5 | 6 | 7 | 8 | 9 | 10 |
|---|---|---|---|---|---|---|---|---|---|---|---|
| $\Omega_k^{\text{fr}}$ | $\mathbb{Z}$ | $\mathbb{Z}_2$ | $\mathbb{Z}_2$ | $\mathbb{Z}_{24}$ | $0$ | $0$ | $\mathbb{Z}_2$ | $\mathbb{Z}_{240}$ | $\mathbb{Z}_2^2$ | $\mathbb{Z}_2^3$ | $\mathbb{Z}_6$ |
| $\Omega_k^{\text{O}}$ | $\mathbb{Z}_2$ | $0$ | $\mathbb{Z}_2$ | $0$ | $\mathbb{Z}_2^2$ | $\mathbb{Z}_2$ | $\mathbb{Z}_2^3$ | $\mathbb{Z}_2$ | $\mathbb{Z}_2^5$ | $\mathbb{Z}_2^3$ | $\mathbb{Z}_2^8$ |
| $\Omega_k^{\text{O}\langle1\rangle}=\Omega_k^{\text{SO}}$ | $\mathbb{Z}$ | $0$ | $0$ | $0$ | $\mathbb{Z}$ | $\mathbb{Z}_2$ | $0$ | $0$ | $\mathbb{Z}^2$ | $\mathbb{Z}_2^2$ | $\mathbb{Z}_2$ |
| $\Omega_k^{\text{O}\langle2\rangle}=\Omega_k^{\text{Spin}}$ | $\mathbb{Z}$ | $\mathbb{Z}_2$ | $\mathbb{Z}_2$ | $0$ | $\mathbb{Z}$ | $0$ | $0$ | $0$ | $\mathbb{Z}^2$ | $\mathbb{Z}_2^2$ | $\mathbb{Z}_2^3$ |
| $\Omega_k^{\text{O}\langle4\rangle}=\Omega_k^{\text{String}}$ | $\mathbb{Z}$ | $\mathbb{Z}_2$ | $\mathbb{Z}_2$ | $\mathbb{Z}_{24}$ | $0$ | $0$ | $\mathbb{Z}_2$ | $0$ | $\mathbb{Z}_2\times\mathbb{Z}$ | $\mathbb{Z}_2^2$ | $\mathbb{Z}_6$ |
| $\Omega_k^{\text{O}\langle8\rangle}=\Omega_k^{\text{Fivebrane}}$ | $\mathbb{Z}$ | $\mathbb{Z}_2$ | $\mathbb{Z}_2$ | $\mathbb{Z}_{24}$ | $0$ | $0$ | $\mathbb{Z}_2$ | $\mathbb{Z}_{240}$ | $?$ | $?$ | $?$ |
| $\Omega_k^{\text{O}\langle9\rangle}=\Omega_k^{\text{2-orientation}}$ | $\mathbb{Z}$ | $\mathbb{Z}_2$ | $\mathbb{Z}_2$ | $\mathbb{Z}_{24}$ | $0$ | $0$ | $\mathbb{Z}_2$ | $\mathbb{Z}_{240}$ | $\mathbb{Z}_2^2$ | $?$ | $?$ |
| $\Omega_k^{\text{O}\langle10\rangle}=\Omega_k^{\text{2-spin}}$ | $\mathbb{Z}$ | $\mathbb{Z}_2$ | $\mathbb{Z}_2$ | $\mathbb{Z}_{24}$ | $0$ | $0$ | $\mathbb{Z}_2$ | $\mathbb{Z}_{240}$ | $\mathbb{Z}_2^2$ | $\mathbb{Z}_2^3$ | $\mathbb{Z}_6$ |
| $\Omega_k^{\text{O}\langle12\rangle}=\Omega_k^{\text{Ninebrane}}$ | $\mathbb{Z}$ | $\mathbb{Z}_2$ | $\mathbb{Z}_2$ | $\mathbb{Z}_{24}$ | $0$ | $0$ | $\mathbb{Z}_2$ | $\mathbb{Z}_{240}$ | $\mathbb{Z}_2^2$ | $\mathbb{Z}_2^3$ | $\mathbb{Z}_6$ |

Table 3.1: Bordism groups $\Omega_k^{\text{O}\langle n\rangle}$ for various dimensions $k$; note that $\Omega_k^{\text{O}\langle2\rangle}=\Omega_k^{\text{O}\langle3\rangle}$, $\Omega_k^{\text{O}\langle4\rangle}=\Omega_k^{\text{O}\langle5\rangle}=\Omega_k^{\text{O}\langle6\rangle}=\Omega_k^{\text{O}\langle7\rangle}$, and $\Omega_k^{\text{O}\langle10\rangle}=\Omega_k^{\text{O}\langle11\rangle}$ by Bott periodicity. We highlight in blue the realisation of the isomorphism (3.13).

$w_1(TM)=0$ and $w_2(TM)=0$. Dropping the latter, we are left the condition for orientability, which corresponds to an $\text{O}\langle1\rangle$-structure. On the other hand, only demanding $w_2(TM)=0$ corresponds to a $\text{pin}^+$ structure, which does not appear in the Whitehead tower – but it is relevant for orientifolds in M-theory.

We enlarge the number of potential candidates for QG by dropping some of the "lower" obstruction trivialisation conditions for any given entry in the Whitehead tower. In particular, there are up to $2^4-1$ such candidates obtained by discarding at least one of the last four of the vanishing constraints on $\frac{1}{240}p_3(TM)$, $\frac{1}{6}p_2(TM)$, $\frac{1}{2}p_1(TM)$, $w_2(TM)$, and $w_1(TM)$ for a ninebrane structure. We do not know the names for these 15 structures, but the above discussion invites their further study. Such and other spaces may also be chosen as the bases of other Whitehead towers, which in turn might lead to increasingly good approximations to QG. That other Whitehead towers could be relevant to identify QG is consistent with the fact that $\text{pin}^+$ structures appear in M-theory. Quite generally, one may expect that potential anomalies in a theory of quantum gravity are the obstructions encoded in some Whitehead tower(s), which in this sense would offer another elegant organisational structure.

## 4    Fluxes and defects

In this section, we first review and develop in Section 4.1 arguments of [4] showing that the cobordism conjecture requires, or "predicts", the existence of magnetic sources charged under a U(1) gauge symmetry, generally understood as defects. These arguments also show that the cobordism conjecture implies the completeness hypothesis as well the absence of global symmetries. Having reviewed these points in Section 4.1.1, we briefly extend the

reasoning to electric sources in Section 4.1.2, allowing to recover otherwise missing branes. We turn in Section 4.2 to an attempt at reformulating the whole discussion in the language of bordism groups. Section 4.2.1 focuses on describing higher U(1)-bundles with connection, while Section 4.2.2 discusses the inclusion of magnetic defects. We finally turn in Section 4.3 to the particular case of a Kaluza–Klein monopole, discussing how to predict this extended stringy object from the cobordism conjecture.

## 4.1 Electromagnetic defects from the cobordism conjecture

### 4.1.1 Magnetic defects and completeness hypothesis: review

Building on [4], we consider in the following closed $k$-dimensional oriented manifolds $M$ and $N$. They are taken as boundaries of a spatial bordism $W$ with orientation: we have $\partial W \cong M \sqcup (-N)$. We consider in addition a U(1) gauge symmetry, with an associated flux $F$. This flux is a $k$-form locally given as $F = \mathrm{d}A$, and $F$ is consistently defined on the bordism $W$ as well as on its boundaries $M$ and $N$. In mathematical terms, the bordism is the base of a (higher) principal U(1)-bundle, and the U(1)-flux $F$ on $W$ admits appropriate restrictions on the boundary. We are interested in the flux number on $M$, defined as $n_M = \int_M F$. Since $M$ is closed and hence compact, $F$ should be quantised, meaning $n_M \in \mathbb{Z}$. Following the notation of Section 2.2, we consider the compactification datum $(M, n_M)$, and we want to construct an equivalence relation and a bordism group for such data, for which the cobordism conjecture (2.14) will be satisfied.

We first restrict ourselves to bordisms on which $\mathrm{d}F = 0$: this information is part, together with the orientation, of the defining information of the bordism. We consider two compactifications on $M$ and $N$ in the same equivalence class, i.e. connected by a bordism $W$. We then have

$$0 = \int_W \mathrm{d}F = \int_{\partial W} F = \int_M F + \int_{-N} F = \int_M F - \int_N F \iff n_M = n_N\,, \qquad (4.1)$$

where the minus sign is due to the orientation. This implies that a bordism group for the datum $(M, n_M)$, with $\mathrm{d}F = 0$, has an infinite number of classes, i.e. one per integer $n_M$, because such bordisms cannot connect two compactifications with different flux numbers. So it does not obey the cobordism conjecture. In particular, the trivial class should carry a vanishing flux number, as being that of the empty set. In other words, to obey the cobordism conjecture with this datum, one needs to restrict to only $n_M = 0$. We now recall from Section 2.3.2 that $\mathrm{d}F = 0$ can be interpreted as having a global U(1) symmetry. The integer $n_M$ can then be understood as a global charge on $M$, and the cobordism conjecture enforces this charge to vanish. This amounts to require the absence of the global symmetry.

The above makes it clear that the flux is preserved in the bordism. One way to cancel it (for the cobordism conjecture to hold) is then to introduce charged sources or monopoles: physically, they are the endpoints of flux lines. In other words, the cobordism conjecture requires the existence of objects, generically called defects, that carry the charge necessary to cancel the flux. More precisely, given $n_M$, we now consider bordisms where $\mathrm{d}F = j_\mathrm{m}$, such that $\int_W j_\mathrm{m} = n_M$. The current $j_\mathrm{m}$ is the (magnetic) source or defect contribution, and $n_M$ is its charge. As explained in Section 2.3.2, this can also be interpreted as gauging the U(1)-symmetry. This new bordism provides the desired flux cancellation as follows:

$$n_M = \int_W j_\mathrm{m} = \int_W \mathrm{d}F = n_M - n_N \iff n_N = 0\,. \qquad (4.2)$$

The compactification $(M, n_M)$ is now in the same equivalence class as a compactification without flux. As explained, this is necessary to be bordant to the empty set, so we are now in a better position to satisfy the cobordism conjecture.

We have to consider bordisms with $dF = j_m$ and $\int_W j_m \in \mathbb{Z}$. A last step is to span fully $\mathbb{Z}$, i.e. include at least one $j_m$ per integer. In other words, we need defects of all possible integer charges. Any flux number can then be bordant to any other flux number, and in particular with 0. This is a necessary condition to obey the cobordism conjecture. We conclude that the cobordism conjecture implies the need of having defects of all possible charges: this is precisely the completeness hypothesis [31, 47], i.e. the charged state spectrum is completely filled with existing objects.

The above was an attempt to devise bordisms and a bordism group that would provide equivalence relations for the datum $(M, n_M)$, and obey the cobordism conjecture. It raises however two questions. First, it would be satisfying to capture the information of the U(1)-bundle, and the related flux numbers, directly in a bordism language. One further step would be to include as well the information of the defects. It is also not entirely clear how to properly include sources of all charges at once, allowing to go from $\mathbb{Z}$ to a trivial group. We tackle these questions in Section 4.2.

Secondly, let us apply the above to type II string theories and their NSNS $H$-flux and RR $F_q$-fluxes. The cobordism conjecture then "predicts" respectively the existence of $NS_5$-branes and $D_p$-branes, as well as the corresponding anti-branes of opposite charge. Let us be more precise on the dimensions: the magnetic source contribution $j_m$ is along the source $(k+1)$-dimensional transverse volume, the volume of $W$, and we do not include the time direction here. The magnetic source is thus along the remaining extended $D - 2$ spatial dimensions. For the $H$-flux, we have $D + k = 10$ and $k = 3$, thus a 5-dimensional magnetic source: the $NS_5$-brane. For $F_q$ with $0 \leqslant q \leqslant 5$, we similarly get $D_p$ with $p = 8 - q$, i.e. $3 \leqslant p \leqslant 8$. The same reasoning can be applied to M-theory and its $G_4$-flux predicts the existence of $M_5$-branes. This raises the question of the remaining branes that we know of: how are those predicted? We now turn to this side question.

### 4.1.2 Electric defects

We have reviewed in Section 4.1.1 how considering compact manifolds with U(1)-fluxes $F$, and satisfying the cobordism conjecture (2.14), leads to considering bordisms where $dF = j_m$, predicting the existence of magnetic sources. We identified in string and M-theory such magnetic defects as certain branes, but also noticed that other branes were not predicted this way. In particular, we are missing in string theory $D_0, D_1, D_2$, and $M_2$-branes in M-theory. Last but not least, the $F_1$ fundamental string is also not predicted. The reason for this is simple: those are rather electric sources for some of the fluxes (and associated gauge fields). Electric sources should be obtained through the relation

$$d *_{D+k} F = j_e \,, \tag{4.3}$$

where $F$ should be along the time direction, and $j_e$ should not. The current $j_e$ is along the transverse (space) volume to the source, so it is at most a $(D + k - 1)$-form. This implies that $F$ is at least a 2-form, i.e. cannot be a 0- or 1-form. The NSNS and RR fluxes are at most 5-forms. $F_5$ in ten dimensions would electrically be sourced by a $D_3$, already obtained as a magnetic source; this is consistent with $F_5$ being anti-self-dual on-shell. So we are left

with $F$ being a 2-, 3-, or 4-form. The known fluxes are enough to give rise to the missing electric sources mentioned above, as one can verify with dimensions: the RR $F_{2,3,4}$ would give $D_{0,1,2}$-branes respectively, the NSNS $H$-flux the $F_1$ string, and in M-theory, the $G_4$-flux would give $M_2$-branes. We still need to justify why satisfying the cobordism conjecture would predict these objects, in particular the need for the equation (4.3).

To obtain such a prediction, one has to consider a compactification resulting in a U(1)-flux $F$, being a $D$-form along the $D$-dimensional extended spacetime. This is a different starting point than for magnetic defects. In particular, since $F$ is now along the time direction, $\int F$ is not on a compact space anymore. However, one can consider the dual flux $\tilde{F} = *_{D+k}F$: this is a $k$-form on $M$. As before, this $\tilde{F}$ then has to be quantised: $\int_M \tilde{F} = \tilde{n}_M \in \mathbb{Z}$. We then proceed as for magnetic sources: satisfying the cobordism conjecture requires, as a necessary condition, to have bordisms $W$ on which $\mathrm{d}\tilde{F} = j_{\mathrm{e}}$, with $\int_W j_{\mathrm{e}}$ spanning all integers. This is needed to build a bordism group for which $[(M, \tilde{n}_M)] = 0$. We recover in this way the need for the relation (4.3). We conclude that the cobordism conjecture predicts the existence of electric defects, whose contribution is given by $j_{\mathrm{e}}$ in (4.3), and one should include all integer charges. This implies again the completeness hypothesis. We recover in particular the branes missing in Section 4.1.1 and corresponding to anti-branes.

## 4.2 Fluxes and defects in bordisms

We now aim at describing the physical setting of Section 4.1.1 in the rigorous language of bordism groups. Beyond the topological structure (in that case orientation), two new physical ingredients require a mathematical description: U(1)-fluxes and defects, which we now tackle in turn. Along the way, we briefly discuss bordism groups for arbitrary geometric structures.

### 4.2.1 Higher bundles with connection

In Section 4.1.1 we discussed compactifications on $k$-dimensional manifolds involving $k$-form U(1)-fluxes in the context of the cobordism conjecture. Mathematically, this involves connections on (higher) U(1)-bundles and their curvatures. Such *geometric* structures are a priori not included in the traditional study of bordism groups reviewed in Section 2.1, where we exclusively considered *topological* structures such as orientations, spin, or $G$-bundles *without* connection. More concretely, to discuss fluxes in the language of bordisms, groups of the form $\Omega_k^\xi(B^n\mathrm{U}(1))$ were used in [17, Sect. 6.3]. This manifestly captures higher U(1)-bundles *without* connection. The first goal of the present section is to give an independent definition of bordism groups $\Omega_k^\xi(\mathbf{B}_\nabla^n\mathrm{U}(1))$ that explicitly also include connections (and hence fluxes), and then to show that indeed[5]

$$\Omega_k^\xi\big(\mathbf{B}_\nabla^n\mathrm{U}(1)\big) \cong \Omega_k^\xi\big(B^n\mathrm{U}(1)\big). \tag{4.4}$$

By setting $n = k - 1$ and $\xi = \mathrm{SO}$, we are in the setting of Section 4.1.1 with $j_{\mathrm{m}} = 0$; the case of non-zero magnetic currents is discussed in Section 4.2.2.

While bordism groups are not sensitive to connections, this is however not true of geometric structures $\mathcal{S}$ in general, such as (pseudo) Riemannian metrics (possibly with constraints on Ricci curvature), complex or symplectic structures. We will define the associated general bordism groups $\Omega_k^\mathcal{S}$ in (4.8) below, which we expect to play a role in connection with the

---

[5] We thank Miguel Montero for helpful discussions on this point.

cobordism conjecture as well. Other than that, in the remainder of the present section we shall explain the ingredients of the left-hand side of (4.4) in more detail; some readers may wish to skip ahead to Section 4.2.2.

First we recall that for a Lie group $G$, the set of isomorphism classes of $G$-bundles over a manifold $M$ is classified up to homotopy by continuous maps $M \longrightarrow BG$. Moreover, for any positive integer $n$, it follows from the discussion around (3.1) and (3.2) that $B^{n-1}\mathrm{U}(1)$-bundles (to which we also refer to as *(higher) circle bundles*, as $\mathrm{U}(1) \cong S^1$) over a manifold $M$ are classified by integral cohomology of $M$:

$$\left[M, B^n\mathrm{U}(1)\right] \cong \left[M, B^{n+1}\mathbb{Z}\right] \cong \left[M, K(\mathbb{Z}, n+1)\right] \cong H^{n+1}(M; \mathbb{Z}). \tag{4.5}$$

Including connections can be described analogously, however at the price of going from the world of topological spaces to that of *stacks*, i.e. simplicial sheaves on the site of smooth manifolds. This is clearly explained e.g. in [48] and [49, Sect. 3], to which we refer for details as well as for more mathematical background. Examples of stacks are already familiar from $\mathrm{U}(1)$-connections in Maxwell theory, or Kalb–Ramond fields for $\mathrm{U}(1)$-2-bundles (also known as bundle gerbes) with connection.

Next we give a brief summary of how stacks describe higher circle bundles with connection. Recall that every manifold $M$ can be equivalently described in terms of the presheaf $\mathcal{F}_M$ which associates to any "test manifold" $U$ the space of smooth maps $U \longrightarrow M$, denoted $\mathrm{C}^\infty(U, M)$. Since manifolds are glued together from local patches, it suffices to consider only test manifolds $U$ which are diffeomorphic to some $\mathbb{R}^n$. Smooth maps $M \longrightarrow N$ then correspond to natural transformations $\mathcal{F}_M \longrightarrow \mathcal{F}_N$. It turns out that $\mathcal{F}_M$ satisfies a certain gluing property which makes it a sheaf. In general, from every presheaf one can canonically construct a sheaf in a process called sheafification (which is left adjoint to the forgetful functor).

Given a Lie group $G$, one can consider the group $\mathrm{C}^\infty(U, G)$ for every test manifold $U$, and turn it into the one-object groupoid $*/\!\!/\mathrm{C}^\infty(U, G)$. Taking the nerve $N$, we get a simplicial presheaf $U \longmapsto N(*/\!\!/\mathrm{C}^\infty(U, G))$, and by definition the stack $\mathbf{B}G$ is its sheafification. The classification of $G$-bundles on $M$ via classifying maps naturally lifts to the statement that the groupoid of $G$-bundles on $M$ is given by certain maps from $M$ (viewed as a stack) to $\mathbf{B}G$, see e.g. [49, Sect. 3.2.1]. Similarly, there is a stack of $G$-bundles with connection $\mathbf{B}_\nabla G$ which is analogously constructed from the action groupoids $\Omega^1(U, \mathfrak{g})/\!\!/\mathrm{C}^\infty(U, G)$, where $\mathfrak{g}$ is the Lie algebra of $G$, and $\Omega^1(-, \mathfrak{g})$ is the sheaf of $\mathfrak{g}$-valued 1-forms. Indeed, there is an equivalence of groupoids between $G$-bundles with connection on $M$ and certain maps from $M$ to $\mathbf{B}_\nabla G$. In other words, the analogue of the classifying space $BG$ for bundles *with* connection is given by the stack $\mathbf{B}_\nabla G$.

If $G$ is an abelian group like $\mathrm{U}(1)$, the delooping procedure $\mathbf{B}$ can be applied any number of times, leading to a stack $\mathbf{B}^n\mathrm{U}(1)$ that classifies $\mathrm{U}(1)$-$n$-bundles (or $(n-1)$-gerbes) for all positive integers $n$. This is reviewed in explicit detail e.g. in [50, Sect. 2.3]; in particular one can extract the local data of $\mathrm{U}(1)$-bundles for $n = 1$, as well as the local data and their cocycle identities of $\mathrm{U}(1)$-bundle gerbes for $n = 2$.

Connections on $\mathrm{U}(1)$-bundles similarly generalise to connections on $\mathrm{U}(1)$-$n$-bundles. They are classified by a stack $\mathbf{B}_\nabla^n\mathrm{U}(1)$ that may be constructed by applying the Dold–Kan correspondence to the Deligne complex

$$\mathrm{C}^\infty(-, \mathrm{U}(1)) \xrightarrow{\;\mathrm{d}\log\;} \Omega^1(-, \mathbb{R}) \xrightarrow{\;\mathrm{d}\;} \cdots \xrightarrow{\;\mathrm{d}\;} \Omega^n(-, \mathbb{R}) \tag{4.6}$$

of presheaves, see e. g. [49, Sect. 2.3.2]. As reviewed in [50, Sect. 3.3], in terms of local data (simplex-wise for the simplicial presheaf associated to (4.6)) this reduces to a description of Maxwell and Kalb–Ramond fields for $n = 1$ and $n = 2$, respectively. In general, unravelling $\mathbf{B}^n_\nabla U(1)$ in local data reveals that the space of connections on a given U(1)-$n$-bundle is an affine space (over differential forms).[6]

We can now state that a generalisation of (4.5) is

$$\left\{ \text{isomorphism classes of U(1)-}n\text{-bundles with connection on } M \right\} \cong \check{H}^{n+1}(M)\,, \qquad (4.7)$$

where the right-hand side is the $(n+1)$-th ordinary differential cohomology of the manifold $M$, see e. g. the reviews [51, Sect. 2], [52, Sect. 5.1–5.2], or [53, Sect. 6.4.16].

Finally, to define bordism groups for arbitrary topological and/or proper geometric structures $\mathcal{S}$, recall that at the end of Section 2.1 we noted that the bordism groups $\Omega^\xi_k$ discussed there are precisely the connected components $\pi_0(\mathrm{Bord}^\xi_{k+1,k})$ of the bordism categories that appear in topological quantum field theory of Atiyah–Segal type. To study more general smooth functorial field theories, in [54, 55] a smooth $(\infty, k)$-category $\mathrm{Bord}^\mathcal{S}_k$ is constructed as a certain presheaf for every stack $\mathcal{S}$. Hence by looping $k - 1$ times (for which the standard notation is $\Omega^{k-1}$, not to be confused with bordism groups or differential forms), i. e. only considering trivial objects and trivial 1- up to $(k - 2)$-morphisms, and then taking isomorphism classes, we obtain a bordism group of $k$-dimensional manifolds with $\mathcal{S}$-structure,

$$\Omega^\mathcal{S}_k := \pi_0\Big(\Omega^{k-1}\big(\mathrm{Bord}^\mathcal{S}_k(\mathrm{pt})\big)\Big). \qquad (4.8)$$

We are not aware of any explicit computations of bordism groups with non-trivial geometric structures that do not form contractible spaces for given underlying topological structures (as in the case of connections). One might expect that such groups are often non-discrete, as opposed to bordism groups for topological structures as e. g. in Table 3.1.

In the case of contractible local moduli spaces of geometric structures $\mathcal{S}$, any two representatives of elements of $\Omega^\mathcal{S}_k$ with the same underlying topological structure are connected by a bordism which topologically is just a cylinder, and whose geometric structure interpolates between the two boundaries. In particular, if $\mathcal{S}$ is some tangential structure $\xi$ together with $\mathbf{B}^n_\nabla U(1)$, i. e. higher circle bundles with connection on $\xi$-manifolds, then the moduli space is an affine space (over the vector space of connections on the trivial higher circle bundle), hence in particular contractible. This in turn means that any two representatives of elements of $\Omega^\xi_k(\mathbf{B}^n_\nabla U(1))$ with the same underlying higher circle bundle (but possibly different connections) are connected by a bordism which topologically is just a cylinder, and whose connection interpolates between the two boundaries. It follows that there is an isomorphism $\Omega^\xi_k(\mathbf{B}^n_\nabla U(1)) \cong \Omega^\xi_k(B^n U(1))$ as in (4.4).

### 4.2.2 Including defects

Having identified the bordism group relevant to higher U(1)-bundles with connection as being $\Omega^\xi_k\big(\mathbf{B}^{k-1}_\nabla U(1)\big)$, which by (4.4) is isomorphic to the group $\Omega^\xi_k\big(B^{k-1}U(1)\big)$ without connection, we now focus on the second physical ingredient that appears in Section 4.1.1: defects. We would like to show that, once we introduce and describe them in a bordism group, the latter becomes closer to a trivial one, thus confirming the intuition from Section 4.1.1.

---

[6]We are grateful to Domenico Fiorenza for helpful discussions and generous explanations of these matters.

Let us recall some notation. We consider a spatial bordism $W$ which serves as the base of a higher U(1)-bundle with $k$-form field strength $F$. The boundary of $W$ is made of two $k$-dimensional oriented compact manifolds $M$ and $N$, carrying flux integers $n_M = \int_M F$ and $n_N = \int_N F$. As argued in Section 4.1.1, having $n_M \neq n_N$ requires a non-closed $F$ on $W$, i.e. the presence of magnetic defects with current $dF = j_\mathrm{m}$. In other words, the presence of defects can be seen through a non-zero $j_\mathrm{m}$.

By definition, a connection on a U(1)-$(k-1)$-bundle over $W$ gives rise to a closed curvature form $F \in \Omega^k(W)$, meaning $dF = 0$. Hence (higher) Maxwell theory with a non-zero magnetic current $j_\mathrm{m}$ cannot be described within the framework of Section 4.2.1. As explained in [56, Sect. 2–3], it is natural to interpret $j_\mathrm{m}$ as an element $\check{j}_\mathrm{m} \in \check{H}^{k+1}(W)$ of differential cohomology. Then the twisted Bianchi identity $dF = j_\mathrm{m}$ can be lifted to a trivialisation of $\check{j}_\mathrm{m}$ by the differential cohomology class $\check{F} \in \check{H}^k(W)$ associated to $F$. Equivalently, one may think of this as a U(1)-$(k-1)$-bundle with connection that is "twisted" by a U(1)-$k$-bundle with connection, see e.g. [53, Sect. 1.2.6–1.2.7].

Below we will provide a more lowbrow description that also has a direct physical interpretation. As preparation, we will first focus on the case $dF = 0 = j_\mathrm{m}$ and discuss a few properties of $\Omega_k^{\mathrm{SO}}(B^{k-1}U(1))$. Here and in the following, we consider $\xi = \mathrm{SO}$ since this is the minimal requirement for our argument to apply.

If $dF = 0$, the flux number is preserved through the bordism as discussed in Section 4.1.1. In particular, a manifold $M$ with flux number $n_M$ is bordant to itself. Hence $(M, n_M)$ and $(M, n_M + 1)$ represent different elements of $\Omega_k^{\mathrm{SO}}(B^{k-1}U(1))$, and there is an infinite number of elements in $\Omega_k^{\mathrm{SO}}(B^{k-1}U(1))$ classified by integers,[7] in other words that

$$\Omega_k^{\mathrm{SO}}(B^{k-1}U(1)) \text{ admits } \mathbb{Z} \text{ as a subgroup for all } k \geqslant 2. \qquad (4.9)$$

A more formal proof for (4.9) is as follows. First recall that the Thom homomorphism $\Omega_*^{\mathrm{SO}}(X) \longrightarrow H_*(X, \mathbb{Z})$ is surjective and that the Hurewicz isomorphism gives $H_k(X) \cong \pi_k(X)$ for any $(k-1)$-connected space $X$, with $k \geqslant 2$. Choose now $X := B^{k-1}U(1) \simeq B^k\mathbb{Z}$. By construction, this is $(k-1)$-connected and we have $\pi_k(B^k\mathbb{Z}) = \pi_k(K(\mathbb{Z}, k)) = \mathbb{Z}$. The statement (4.9) follows since group homomorphisms preserve subgroups.

Consider now $M$ and $N$ carrying the same flux integers $n_M = n_N$. In this case, whether or not the two compactifications are bordant in $\Omega_k^{\mathrm{SO}}(B^{k-1}U(1))$ seems independent of the gauge bundle information, and relies only on $M$ and $N$. One may then ask whether they are in the same class in $\Omega_k^{\mathrm{SO}}$. The latter is however not always trivial (see Table 2.1). This reasoning could imply the following equality

$$\Omega_k^{\mathrm{SO}}(B^{k-1}U(1)) \stackrel{?}{=} \Omega_k^{\mathrm{SO}} \times \mathbb{Z} \quad \text{for all } k \geqslant 2. \qquad (4.10)$$

We note in particular that for $k = 2$, $\Omega_2^{\mathrm{SO}}(BU(1)) = \mathbb{Z}$, see e.g. [22, Sect. 3.1.4], while $\Omega_2^{\mathrm{SO}} = 0$. In the discussion below, we will find an interpretation of the quotient $\Omega_k^{\mathrm{SO}}(B^{k-1}U(1))/\mathbb{Z}$.

We now turn to the case of magnetic defects, i.e. $dF = j_\mathrm{m}$, aiming at formulating it in terms of a bordism group. For this, let $\Omega_k^{j_\mathrm{m}}$ denote the bordism group whose elements have the same representatives as $\Omega_k^{\mathrm{SO}}(B^{k-1}U(1))$, but where bordisms $W$ come with U(1)-$(k-1)$-bundles together with $k$-forms $F$ which satisfy $\int_W dF = n \cdot n_j$ for a fixed integer $n_j$ (that we

---

[7]As a side remark, we recall from Section 2.3.2 that $dF = 0$ can be interpreted as the existence of a global symmetry. We see then from (4.9), where the bordism group is not trivial, that it is incompatible with the cobordism conjecture. This will be (partially) resolved by introducing defects, which precisely break the symmetry.

interpret as $\int_W j_\mathrm{m}$) and an arbitrary integer $n$. Put differently, $n_j$ is the minimal non-zero magnetic charge. We first discuss a property of this bordism group, similar to the discussion around (4.9). For this, we first consider a bordism $W$ such that $\int_W \mathrm{d}F = n_j$, and $\partial W$ consists of two copies of some $k$-manifold $M$, each carrying respective flux numbers $n_M$ and $n'_M$. As argued in Section 4.1.1, such a bordism would impose $n'_M = n_M + n_j$. This can be iterated, leading to the conclusion that $(M, n'_M)$ is bordant to $(M, n_M)$ for $n'_M = n_M \bmod n_j$. In other words, only $n_j$ such elements of $\Omega_k^{j_\mathrm{m}}$ can differ. This argument leads us to propose the claim

$$\Omega_k^{j_\mathrm{m}} \text{ admits } \mathbb{Z}_{n_j} \text{ as a subgroup for all } k \geqslant 2. \qquad (4.11)$$

Physically, this means that introducing $n_j$ defects reduces the previous $\mathbb{Z}$ subgroup to $\mathbb{Z}_{n_j}$. The two extreme cases are $n_j = 0$ where $\mathbb{Z}_0 = \mathbb{Z}$ and we are back to the case without defect, and $n_j = 1$ where $\mathbb{Z}_1 = 0$, i.e. $M$ with any flux number is bordant to any other and the subgroup gets trivialised.

To prove the claim (4.11) and more generally compute the bordism group $\Omega_k^{j_\mathrm{m}}$ corresponding to the case with defects, we could work in the formalism of twisted higher U(1)-bundles mentioned above. Instead, in the following we will present an alternative, more pedestrian approach to the relevant bordism groups.

We now restrict to the common situation of localised sources: by this we mean that $j_\mathrm{m}$ only has compact support in the interior of $W$, one possibility being that it is proportional to a (sum of) $\delta$-function(s). Describing such sources is not necessarily easy given the possible associated singularities. However, as noticed e.g. in [4,17,56], one may trade this description of a charged defect for a sphere with certain units of flux thanks to Gauss' law. By the support condition on $j_\mathrm{m}$, it is non-zero only in a $(k+1)$-ball $B_{k+1}$ with $\partial B_{k+1} = S^k$ in the interior of $W$. We then have

$$n_j = \int_W j_\mathrm{m} = \int_{B_{k+1}} j_\mathrm{m} = \int_{B_{k+1}} \mathrm{d}F = \int_{S^k} F, \qquad (4.12)$$

where the $S^k$ carries units of flux that are equal to the charge $n_j$ of $j_\mathrm{m}$. We are now going to make use of this idea to express the presence of a magnetic defect in a bordism language.

We define another bordism where we remove the ball: $W' = W \setminus B_{k+1}$. One has the same $F$ everywhere on $W'$ as on $W$, so we use the same symbol $F$ for both. Because $j_\mathrm{m}$ had support in the ball, one has $\mathrm{d}F = 0$ on $W'$. The bordism $W'$ is thus convenient: we have traded the situation of a bordism with defects for a bordism without any defects, that we described previously.[8] The statement is that *the bordism $W$ carrying a localised $j_\mathrm{m}$ is equivalent to another bordism $W'$ without $j_\mathrm{m}$, but with an additional spherical boundary carrying corresponding units of flux*. In other words, the pairs $(M, n_M)$ and $(N, n_N)$ being bordant in $\Omega_k^{j_\mathrm{m}}$ is equivalent to $(M, n_M) \sqcup -(N, n_N)$ being bordant to $(S^k, n_j)$ in $\Omega_k^{\mathrm{SO}}(B^{k-1}\mathrm{U}(1))$. We can verify directly on $W'$ the compensation of units of flux on its boundary

$$0 = \int_{W'} \mathrm{d}F = \int_{\partial W'} F = \int_M F - \int_N F - \int_{S^k} F \iff n_j = \int_{S^k} F = n_M - n_N, \qquad (4.13)$$

---

[8]One might be worried of facing $\mathrm{d}F = 0$ on $W'$, that can be viewed as having a global symmetry. But the latter can be interpreted as an accidental symmetry in an effective description. Indeed, viewing the source as a (fluxed) sphere is precisely an effective description of it, instead of a UV, fundamental, one. Quantum gravity actually considers the (localised) source, which rather leads to gauging the symmetry (see Section 2.3.2).

and we illustrate the equivalent description as follows:

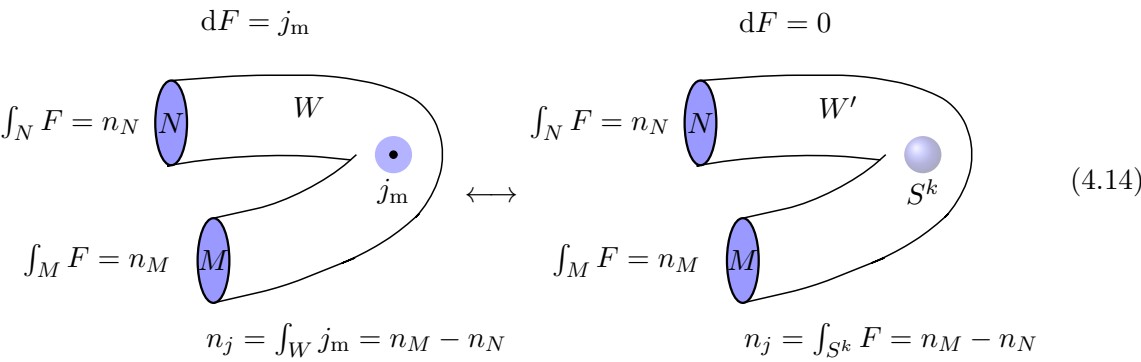

$$n_j = \int_W j_{\mathrm{m}} = n_M - n_N \qquad\qquad n_j = \int_{S^k} F = n_M - n_N$$

(4.14)

We argued that in $\Omega_k^{\mathrm{SO}}(B^{k-1}\mathrm{U}(1))$, manifolds carrying different flux numbers are not bordant. For a given $n_j$, the pairs $(S^k, n \cdot n_j)$ for any integer $n$ are therefore not bordant in $\Omega_k^{\mathrm{SO}}(B^{k-1}\mathrm{U}(1))$. $(S^k, n_j)$ thus generates a subgroup $n_j\,\mathbb{Z}$. Physically, these pairs correspond to having multiple times $n_j$ charged defects. To satisfy the cobordism conjecture (and in line the completeness hypothesis), we know that defects of all charges should be included at once, meaning that all these pairs have to be identified. The latter amounts algebraically to quotienting by $n_j\,\mathbb{Z}$, i.e.

$$\Omega_k^{\mathrm{SO}}(B^{k-1}\mathrm{U}(1)) \,/\, (n_j\,\mathbb{Z})\,. \tag{4.15}$$

We showed in addition the property (4.9), with the $\mathbb{Z}$ subgroup related to flux numbers. The quotient (4.15) therefore provides a $\mathbb{Z}_{n_j}$ subgroup.

This is certainly reminiscent to the $\mathbb{Z}_{n_j}$ appearing in (4.11): this is not an accident. The relation can be understood thanks to the equivalence described above with (4.14), and the fact that $(S^k, n \cdot n_j)$ and $(S^k, n_j)$ are already bordant, i.e. identified, in $\Omega_k^{j_{\mathrm{m}}}$ as argued around (4.11). So the case of multiple charges is already included when considering a single $n_j$ in $\Omega_k^{j_{\mathrm{m}}}$. This leads us to propose the following isomorphism, followed by a proposed equality from (4.10):

$$\Omega_k^{j_{\mathrm{m}}} \;\cong\; \Omega_k^{\mathrm{SO}}(B^{k-1}\mathrm{U}(1)) \,/\, (n_j\,\mathbb{Z}) \;\overset{?}{=}\; \Omega_k^{\mathrm{SO}} \times \mathbb{Z}_{n_j}\,. \tag{4.16}$$

In case $n_j = 1$, this includes a single magnetic defect or the "fundamental" charge, as well as all its multiples. Having $n_j = 1$ could make the quotient (4.15) trivial since $\mathbb{Z}_1 = 0$. Let us emphasise that this is in agreement with the cobordism conjecture, since including all defects through this quotient (or with the appropriate $\Omega_k^{j_{\mathrm{m}}}$) trivialises the bordism group. If it is not trivial, this is not due to fluxes and defects but rather to the geometry of the compact space, as indicated by the factor in the right-hand side of (4.16). We argued that settling whether the quotient is trivial or not amounts to determining whether the pair $(S^k, 1)$ is a unique generator in $\Omega_k^{\mathrm{SO}}(B^{k-1}\mathrm{U}(1))$, or whether there are more.

## 4.3 The Kaluza–Klein monopole problem

In previous sections, we discussed how the cobordism conjecture can "predict" the existence of some branes in string theory. These branes shared the property of sourcing supergravity RR or NSNS U(1)-fluxes. Here, we are interested in a somewhat different extended object that exists in 10-dimensional string theory: the Kaluza–Klein monopole, $KKm$ [57,58]. This

6-dimensional NSNS object is the T-dual to an $NS_5$-brane. Contrary to the latter that (magnetically) sources an NSNS $H$-flux, the $KKm$ has the particularity to be a purely gravitational solution. Indeed, as a supergravity solution, the $KKm$ is given only in terms of a metric, and no RR or NSNS gauge field. We provide more details and references below and in Appendix A.

The absence of RR or NSNS U(1)-gauge flux makes it at first sight less obvious how to apply the reasoning of the previous sections, that would predict a stringy object as a defect needed to cancel a flux number. In addition, because of the prime role played by the metric in the $KKm$, one may consider bordism groups with pseudo Riemannian geometric structure, mentioned at the end of Section 4.2.1, as relevant to predict these stringy objects. This leads us to "the Kaluza–Klein monopole problem":

$$\textit{How does the cobordism conjecture predict the Kaluza–Klein monopole?} \qquad (4.17)$$

In the following, we argue in favour of this answer:

$$\textit{Kaluza–Klein monopoles can serve as defects to kill } \Omega_2^{\mathrm{SO}}(B\mathrm{U}(1)). \qquad (4.18)$$

Let us start from the $KKm$ as a 10-dimensional supergravity solution. It is then only given by the metric (see e.g. [59, (3.42)])

$$\mathrm{d}s^2 = \mathrm{d}s_6^2 + f\,\mathrm{d}s_3^2 + f^{-1}\,(\mathrm{d}x + a\,\mathrm{d}y)^2\,, \text{ where } f = e^{\phi_K} - \frac{q_K}{\rho}\,, \qquad (4.19)$$

$$\text{and } \mathrm{d}s_3^2 = \mathrm{d}\rho^2 + \rho^2\mathrm{d}\varphi^2 + \rho^2\sin^2\varphi\,\mathrm{d}y^2\,, \ a = \cos\varphi\,\rho^2\partial_\rho f \ \overset{\rho>0}{=\!=\!=} \ q_K\cos\varphi\,.$$

The 6-dimensional metric $\mathrm{d}s_6^2$ stands for the Minkowski one, and the 3-dimensional metric $\mathrm{d}s_3^2$ is the metric of the Euclidian flat space, expressed here in spherical coordinates $\{\rho, \varphi, y\}$. The warp factor $f$ depends on the constants $\phi_K$ and $q_K$. Finally, the crucial ingredient of the $KKm$ is the line along $\mathrm{d}x$, which is fibred over the 3-dimensional base with metric $\mathrm{d}s_3^2$, via the connection 1-form $a\,\mathrm{d}y$. In the following, we will consider this line to be a circle with angle $x$, giving us a U(1)-bundle. Physically, the interpretation of this metric is that the $KKm$ is a 6-dimensional extended stringy object (its 7-dimensional world-volume is along the above 6-dimensional Minkowski space and the circle direction along $\mathrm{d}x$), i.e. a codimension-3 object with transverse space along $\{\rho, \varphi, y\}$.

The connection 1-form $a\,\mathrm{d}y$ of the twisted U(1)-bundle can actually serve as the 1-form potential of a magnetic field, the latter being related to the gradient of the warp factor $f$ [58, 60]. The singular locus $\rho = 0$ then hosts a magnetic monopole. In other words, the standard Bianchi identity for magnetic monopoles

$$\mathrm{d}F_2^K = j_{KKm} \qquad (4.20)$$

holds, where $F_2^K$ is the U(1)-field strength 2-form and $j_{KKm}$ is the 3-form current of the magnetic source (see also [39, Sect. 5.1]). In the case of the $KKm$ as given in (4.19), $j_{KKm}$ lives on the 3-dimensional transverse space and is proportional to a $\delta$-function, while the left-hand side of (4.20) is proportional to the 3-dimensional Laplacian of $f$, which is the appropriate Green function.[9]

---

[9]For the interested reader, we add that equivalent expressions of the Bianchi identity (4.20) have been given, in terms of the so-called geometric fluxes $f^a{}_{bc}$, or in terms of the Riemann tensor and the Riemann–Bianchi identity: see [59, (1.11) & (3.5) & (3.107)].

We now consider the same 10-dimensional setting, except for the three transverse dimensions to the $KKm$ which we make compact. More precisely, we consider a 3-dimensional oriented compact bordism $W$ whose boundary is made of two closed 2-dimensional oriented manifolds $M$ and $N$. The $KKm$ is localised at a point on $W$ away from this boundary. The base of the twisted U(1)-bundle is now $W$, with field strength $F_2^K$. We then assume that the Bianchi identity (4.20) holds on $W$, even though finding a concrete solution corresponding to this compact setting remains challenging. This bordism is now equivalent to the one discussed in Section 4.1.1, and we can draw results from there. It is clear that if $\int_M F_2^K \neq 0$, this flux number can be killed by adjusting $\int_W j_{KKm}$, as done for instance in (4.2); we repeat those equalities below in (4.21). The number $\int_W j_{KKm}$ can be interpreted as the charge of $KKm$, proportional to the amount of $KKm$ localised on $W$.

Manifolds connected by bordisms as the one just described represent the same element of the group $\Omega_2^{\mathrm{SO}}(B\mathrm{U}(1))$, as discussed in Section 4.2. The claim is therefore that $KKm$ are the required defects to kill this group, as mentioned in (4.18). We note that $D_6$-branes would also kill this group. The difference is that here, the U(1)-bundle is present in the 10-dimensional spacetime geometry; this is not the case for $D_6$-branes, which are the magnetic monopoles for RR $F_2$-fluxes.

Interestingly, one finds the following properties for this bordism group: $\Omega_2^{\mathrm{SO}}(B\mathrm{U}(1)) = \mathbb{Z}$ and its bordism invariant is the first Chern class $c_1$, see e.g. [22, Sect. 3.1.4], which for any given U(1)-bundle is represented by the curvature 2-form $F_2$ of some connection. In addition, we consider here a 2-dimensional closed manifold $M$, and have $\int_M F_2 \in \mathbb{Z}$ by U(1)-flux quantisation, and these flux numbers may be identified with the elements of $\Omega_2^{\mathrm{SO}}(B\mathrm{U}(1)) = \mathbb{Z}$. This is consistent with having the first Chern class a bordism invariant. We summarise the situation for two manifolds $M, N$ as follows:

$$
\mathbb{Z} \ni n_M - n_N = \int_M F_2^K - \int_N F_2^K = \int_{\partial W} F_2^K = \int_W \mathrm{d}F_2^K = \int_W j_{KKm} \,. \tag{4.21}
$$

Adjusting the number of $KKm$ allows to kill the flux number on the boundary of the bordism, and equivalently the first Chern class of the U(1)-bundle.

We end with a final comment on T-duality. As recalled in Appendix A, the $KKm$ is T-dual to the $NS_5$-brane but also to the $Q$-brane, the latter being a peculiar codimension 2 object. As discussed in previous sections, the $NS_5$-brane can be the defect that kills the group $\Omega_3^{\mathrm{SO}}(B^2\mathrm{U}(1))$. We have just argued that the $KKm$ would kill the group $\Omega_2^{\mathrm{SO}}(B^1\mathrm{U}(1))$, so it is tempting to conjecture that the $Q$-brane would kill $\Omega_1^{\mathrm{SO}}(B^0\mathrm{U}(1))$. Similarly, $D_p$-branes of various $p$ are T-dual to each other, and as discussed in previous sections, they can be the defects killing $\Omega_k^{\mathrm{SO}}(B^{k-1}\mathrm{U}(1))$ for $k = 8 - p$. Based on these comments, it is tempting to think that not only the relevant bordism group is always $\Omega_k^{\mathrm{SO}}(B^{k-1}\mathrm{U}(1))$ for an $(8-k)$-dimensional object, but in addition that this group is preserved through T-duality, meaning that $\Omega_k^{\mathrm{SO}}(B^{k-1}\mathrm{U}(1)) = \Omega_{k-1}^{\mathrm{SO}}(B^{k-2}\mathrm{U}(1))$, etc. Such equalities are a priori not obvious as discussed in Section 4.2.2. Let us nevertheless discuss here this possibility for the three $NS$-branes mentioned. We know that $\Omega_2^{\mathrm{SO}}(B^1\mathrm{U}(1)) = \mathbb{Z}$, and also that $\Omega_1^{\mathrm{SO}}(B^0\mathrm{U}(1)) \equiv \Omega_1^{\mathrm{SO}}(\mathrm{U}(1)) = H_1(S^1) = \mathbb{Z}$ [61, p. 247]. We do not know $\Omega_3^{\mathrm{SO}}(B^2\mathrm{U}(1))$, but we note from Table 2.1 that $\Omega_3^{\mathrm{SO}} = 0$. Therefore, following the argument leading to (4.10), one would conclude on $\Omega_3^{\mathrm{SO}}(B^2\mathrm{U}(1))$ being also $\mathbb{Z}$. The relevant group would then be preserved under T-duality.

# 5 Gravity decoupling limit of the cobordism conjecture

Swampland criteria aim at characterising effective field theories of a quantum gravity theory, in a $D$-dimensional spacetime with $D > 2$. They indicate in particular whether or not a given field theory can be coupled consistently to quantum gravity. In some swampland criteria, the coupling to gravity is seen explicitly through a dependence on the $D$-dimensional Planck mass $M_P$. When $M_P \to \infty$, gravity is decoupled and the constraint becomes either trivially satisfied, empty of content, or violated.[10] This is consistent since swampland criteria are not meant to constrain field theories without gravity. In this section, we discuss the gravity decoupling limit in the case of the cobordism conjecture (2.14).

The formulation of the cobordism conjecture in [4] relies on classical spacetimes that split into a $D$-dimensional non-compact external spacetime and a compact $k$-dimensional Riemannian space $M$. Compactness is central in the mathematical definition of bordisms and bordism groups as presented in Section 2.1. Compactness also played an important role when considering defects in Section 4, for instance in integrals appearing in Section 4.1.1. When considering a string compactification on such a $(D + k)$-dimensional spacetime, the $D$-dimensional Planck mass satisfies

$$M_P^{D-2} \propto \frac{\mathrm{vol}_k}{g_s^2} \,, \tag{5.1}$$

where $\mathrm{vol}_k$ is the volume of $M$ and $g_s$ is the string coupling (see below for a derivation in an example).[11] In a classical regime as considered here, one typically takes $g_s \ll 1$ (see however [5] for less perturbative considerations). If we fix $g_s$ to such a value and send $\mathrm{vol}_k \to \infty$, we deduce from (5.1) that gravity decouples. However, precisely in this limit of infinite volume, we loose compactness, which was just argued to be necessary to the present formulation of the cobordism conjecture. In other words, considering compactness with a finite $g_s$ amounts to have a coupling to quantum gravity. Unless the definition of bordisms and bordism groups can be extended to non-compact spaces, which goes beyond the present framework, the statement of the cobordism conjecture thus appears empty of content in the decoupling limit $\mathrm{vol}_k \to \infty$.

We discuss in the following another possible decoupling limit: we propose to maintain compactness, i.e. a finite $\mathrm{vol}_k$, and send $g_s \to 0$. What happens to the cobordism conjecture in this limit? To address such a question, we use the effective description of string theory in a classical regime provided by a 10-dimensional two-derivatives supergravity. For type II superstrings, the (bosonic) effective action is schematically given by

$$\mathcal{S} = \frac{1}{2\kappa_{10}^2} \int \mathrm{d}^k y \sqrt{|g_k|} \int \mathrm{d}^D x \sqrt{|g_D|} \, \mathrm{e}^{-2\phi} \left[ L_{\mathrm{NSNS}} + L_{\mathrm{NSb.}} + \mathrm{e}^{2\phi} L_{\mathrm{RR}} + \mathrm{e}^{\phi} L_{\mathrm{DBI}} \right], \tag{5.2}$$

with $D + k = 10$, and where the dilaton dependence $\mathrm{e}^{\phi}$ of each contribution (NSNS and RR bulk supergravity, $NS$-branes world-volume contributions, $D_p$-branes and orientifolds DBI contributions) was made explicit. None of the terms $L_{\mathrm{NSNS}}, L_{\mathrm{NSb.}}, L_{\mathrm{RR}}, L_{\mathrm{DBI}}$ contain such an exponential factor.[12] To the action (5.2), one should add the standard topological terms,

---

[10] We thank Miguel Montero for related useful exchanges.

[11] Formula (5.1) illustrates that the arguments of this section do not apply to $D \leqslant 2$.

[12] These terms are schematically given by $L_{\mathrm{NSNS}} = \mathcal{R}_{10} + \ldots$, $L_{\mathrm{RR}} = -\frac{1}{2} \sum_q |F_q|^2 + \ldots$ and $L_{\mathrm{NSb.}}$ as well as $L_{\mathrm{DBI}}$ are given by a volume, a current localising the source and its tension depending purely on the string length $l_s$; we refer to [62, App. A] for complete expressions and conventions.

independent of the metric and the dilaton, as well as fermionic terms. None of those will play a role in the following so we disregard them here.

In a compactification to $D$ dimensions, one considers background-valued fields, that we denote with a superscript 0, and fluctuations around them. One introduces the vacuum expectation value of the dilaton $\phi$, identified with the string coupling constant $\mathrm{e}^{\phi^0} = g_{\mathrm{s}}$, and $\mathrm{e}^{\phi} = g_{\mathrm{s}}\mathrm{e}^{\delta\phi}$. Similarly, the variation of the $k$-dimensional volume around its background value can be introduced, denoted $v$. Following this procedure (see e.g. [63]), an effective theory around a background can then be constructed, whose action and Planck mass $M_{\mathrm{P}}$ are obtained from (5.2) as

$$
M_{\mathrm{P}}^{D-2} = \frac{\int \mathrm{d}^k y \sqrt{|g_k^0|}}{\kappa_{10}^2\, g_{\mathrm{s}}^2}\,, \tag{5.3}
$$
$$
\mathcal{S} = M_{\mathrm{P}}^{D-2} \int \mathrm{d}^D x \sqrt{|g_D|}\, \mathrm{e}^{-2\delta\phi} v\, \frac{1}{2} \left[ \mathcal{R}_D + \tilde{L}_{\mathrm{NSNS}} + L_{\mathrm{NSb.}} + g_{\mathrm{s}}^2 \tilde{L}_{\mathrm{RR}} + g_{\mathrm{s}} \tilde{L}_{\mathrm{DBI}} \right],
$$

where $\tilde{L}_{\mathrm{NSNS}} = -\mathcal{R}_D + L_{\mathrm{NSNS}}$, $\tilde{L}_{\mathrm{RR}} = \mathrm{e}^{2\delta\phi} L_{\mathrm{RR}}$ and $\tilde{L}_{\mathrm{DBI}} = \mathrm{e}^{\delta\phi} L_{\mathrm{DBI}}$. This agrees with the formula (5.1) for $M_{\mathrm{P}}$. The fluctuations $\mathrm{e}^{-2\delta\phi}v$ are then typically absorbed by going to the Einstein frame and do not play a role here.[13] Thanks to this schematic derivation, we see the dependence on $g_{\mathrm{s}}$ in such a regime. We conclude that keeping a finite volume $\mathrm{vol}_k = \int \mathrm{d}^k y \sqrt{|g_k^0|}$ and sending $g_{\mathrm{s}} \to 0$, *the dominant contribution is that of the NSNS sector alone.*

We now speculate that there exists a bordism group $\Omega_k^{\mathrm{QG}_{g_{\mathrm{s}}\to 0}}$, which results from $\Omega_k^{\mathrm{QG}}$ when taking the gravity decoupling limit with a compact space and $g_{\mathrm{s}} \to 0$. Determining $\Omega_k^{\mathrm{QG}_{g_{\mathrm{s}}\to 0}}$ would settle the fate of the cobordism conjecture in this specific gravity decoupling limit: if the cobordism conjecture is trivially satisfied, then $\Omega_k^{\mathrm{QG}_{g_{\mathrm{s}}\to 0}} = 0$, while if it is violated, $\Omega_k^{\mathrm{QG}_{g_{\mathrm{s}}\to 0}} \neq 0$. The previous discussion could help in this purpose. Indeed, the claim is that the representatives of the element(s) of $\Omega_k^{\mathrm{QG}_{g_{\mathrm{s}}\to 0}}$ are only classical pure NSNS backgrounds (including contributions of *NS*-branes). The structure $\mathrm{QG}_{g_{\mathrm{s}}\to 0}$ would correspond to these backgrounds, and it should have a mathematical description in terms of a geometric structure: it would include a pseudo Riemannian structure for the metric, a higher U(1)-bundle for the Kalb-Ramond field, and further structure corresponding to *NS*-branes. Computing the bordism group for such a geometric structure is however challenging. In the remainder of this section, we discuss a few examples of NSNS backgrounds and how they appear as representatives of elements of bordism groups, possibly getting this way a hint on $\Omega_k^{\mathrm{QG}_{g_{\mathrm{s}}\to 0}}$.

Among pure NSNS classical backgrounds with a $k$-dimensional compact space, one obvious subset is that of Ricci-flat backgrounds made of a $D$-dimensional Minkowski spacetime, together with a $k$-dimensional Ricci-flat compact manifold. One can then ask whether all Ricci-flat compact manifolds of dimension $k$ would be bordant in $\Omega_k^{\mathrm{QG}_{g_{\mathrm{s}}\to 0}}$. A partial answer could come from the recent proposal in [64, 65]. The latter claims that Ricci-flat non-supersymmetric compactifications to Minkowski space are unstable. This was verified for 3-dimensional Ricci-flat compact spaces, with instabilities related to bubbles of nothing. This

---

[13]Our discussion assumes that we stay within the regime considered, so in particular, that the fluctuations remain under control.

means that 3-dimensional non-supersymmetric Ricci-flat compact manifolds are bordant to the empty set, within a suitable bordism group.

Another interesting example involving purely NSNS ingredients is the one pointed out in [17, Footnote 51]: $\Omega_4^{\mathrm{Spin}} = \mathbb{Z}$, generated by $K3$, a Ricci-flat manifold. This could be a potential counterexample to having all Ricci-flat compact manifolds bordant in a relevant bordism group. However, an idea is to make this bordism group trivial thanks to some defects [4, 17]. And indeed, the gauge field $B_6$, dual to the $B$-field and electrically sourced by an $NS_5$-brane, is mentioned in [4, Sect. 4.2.1] to play a role in killing the group $\Omega_4^{\mathrm{Spin}}$ in heterotic string theory. Extrapolating from this example, the pure NSNS backgrounds made of Ricci-flat compact manifolds, dressed up with backreacted $NS$-branes, could be bordant to the empty set in some relevant bordism group.

# 6 Summary and outlook

In this paper, we investigated the cobordism conjecture along several directions, with particular focus on its mathematical formulation. After reviewing the conjecture itself in Section 2, together with the necessary mathematical background and some aspects concerning its relation to the role of global symmetries in quantum gravity, in Section 3 we proposed to use the Whitehead tower construction as an organising principle for the topological structures entering the definition of bordism groups. Even if this proposal may not lead to a definite identification of the unknown QG-structure, we observed that it can still point us in the right direction, as bordism groups are typically getting smaller when climbing the tower. Furthermore, in the related Appendix A, we comment briefly on a stringy interpretation of the higher brane structures $O\langle 9 \rangle$ and $O\langle 10 \rangle$ appearing in the Whitehead tower. In Section 4, we concentrated then on the role of fluxes and defects and on how to incorporate them in the language of bordisms. As for fluxes, after giving an intuitive picture in terms of flux numbers, we provided a mathematically rigorous definition of bordism groups of (higher) bundles with connection as well as more general geometric structures. In the case of bundles with (the geometric structure given by a) connection the bordism group is isomorphic to the one with the geometric structure discarded, but this is not expected to be true in general, e.g. in the case of metrics. As for defects, we gave a prescription on how to describe them within a given bordism by cutting out spheres in an appropriate manner. We also pointed out an interesting problem, namely how the cobordism conjecture is capable of predicting Kaluza–Klein monopoles as defects, and we proposed an answer. Finally, in Section 5 we concentrated on how the information on the Planck mass (and thus quantum gravity) is encoded into the conjecture and what happens in the decoupling limit.

We conclude with an outlook. Among all string theory backgrounds, two that are dual to one another certainly share a peculiar relation. Is this specificity somehow manifest when considering bordisms between two such string compactifications? More concretely, one may consider a standard example of two T-dual backgrounds: the first one includes a 3-torus carrying a non-zero, constant, $H$-flux, and the second one includes the T-dual configuration made of a 3-dimensional nilmanifold carrying no $H$-flux. Details and references can be found e.g. in [66] or [67, App. B]. One may wonder whether there exists a bordism group in which the two T-dual compactifications of this example are in the same class, and whether their T-duality relation is of any relevance in this context.[14] Because of the $H$-flux, the bordism group

---

[14] One could contrast this situation by considering other compactifications that are very similar but not

discussed in Section 4.2.1 could be of importance here, and help answering the question (see also the end of Section 4.3). However, one may need to consider extra geometric structure, namely a pseudo Riemannian metric, because of the role played by it in T-duality, unless only topological data eventually matter in characterising these T-dual backgrounds. It is not obvious to us how to construct an explicit 4-dimensional bordism with $H$-flux that would interpolate between the two previously mentioned compactifications. What relevant bordism group would have two T-dual backgrounds as representatives of the same element hence remains unknown.

Let us sketch another idea regarding such a bordism group. Various formalisms have been developed in the last decade to describe backgrounds which are T- or U-dual: these are generalised geometry, double field theory, and exceptional versions of those (see e.g. [71,72]). In these formalisms, one typically considers double or exceptional spaces. Those are made of the physical dimensions, together with other extra dimensions, in such a way that the duality group $G$ provides a $G$-structure to the space. Since the typical T- or U-duality groups are $\mathrm{O}(n,n)$ or $\mathrm{E}_{n(n)}$, one could be led to consider bordism groups $\Omega_k^{\mathrm{O}(n,n)}$ or $\Omega_k^{\mathrm{E}_{n(n)}}$, where $k$ would be the dimension of the double or exceptional space, usually related to the dimension of a representation of the duality groups. The notion of bordism itself might need an extension, since what we used to consider as its boundary may not be smooth manifolds anymore, but rather these spaces with non-physical dimensions as described by double or exceptional geometries (for example the T-folds mentioned in Appendix A). Such tentative bordism groups could be relevant to describe the T-dual compactifications discussed above,[15] but also the T-dual branes mentioned in Appendix A. For those, a unified picture of all branes in the larger exceptional space can be found for instance in Table 2 of [73].

Next we briefly try to connect the cobordism conjecture to the finiteness conjecture [1]. The finiteness conjecture is difficult to state in a precise manner, but it is motivated by a collection of seemingly related ideas. The conjecture is essentially the claim that the number of string theory vacua, up to moduli deformations, is finite. Instead of trying to make this general idea more precise, let us mention a few related pieces of the literature. The first one is the conjecture of [74,75], which states that the number of phenomenologically interesting string vacua is finite, where "phenomenologically interesting" means similar to our universe, in a sense given in those papers. Such a criterion allows to avoid possible counterexamples such as Freund–Rubin solutions. A second, 30-year-old conjecture is that the number of distinct Hodge diamonds of Calabi–Yau 3-folds ($\mathrm{CY}_3$) is finite. This may be extended to $\mathrm{CY}_n$ for all $n$, which is known to be true for $n \in \{1,2\}$. Some evidence for such a claim is related to the observation, in large databases of $\mathrm{CY}_3$, that most of them admit an elliptic fibration, while in [77] it is proven that the number of elliptically fibred $\mathrm{CY}_3$ is finite. A last line of thought regarding finiteness consists in finding upper bounds on certain central charges [78] (see also [1,79]), or on the rank of the allowed gauge groups [14,80,81], thus restricting the possible gauge groups and field content in quantum gravity effective theories.

---

T-dual. An example is a Minkowski solution with a Ricci-flat (non-nilpotent) solvmanifold, that was shown, from various perspectives, to be non-T-dual to the above two backgrounds [68–70]. It would also be interesting to study whether the compact space and flux appearing there can be seen as bordant to the previous compactifications.

[15]In the previous example of the two T-dual compactifications, adding a single non-physical compact dimension to the three compact ones is enough to describe the T-duality in a doubled formalism. Then we cannot refrain from noticing that $\Omega_4^{\mathrm{SO}} = \mathbb{Z}$, which might provide an appropriate bordism group: the $\mathbb{Z}$ could correspond to the toroidal $H$-flux integer.

Finally, recent works on the topic, including [82–85], have provided further interesting insights and formulations of the finiteness conjecture.

Finally we consider the combination of the finiteness conjecture and the cobordism conjecture, and try to formulate the former in the language of bordisms. For the sake of the argument, let us assume that we know what the QG-structure is and that we can compute $\Omega_k^{\text{QG}}$, which will involve information on both topology and geometry. Assuming the cobordism conjecture to be true, this group is given by the trivial class $0 = [\varnothing]$ only. Two situations can now occur: the number of representative of this class may be finite or infinite. In the former case, the finiteness conjecture follows as the statement that the number of null-bordant QG-backgrounds is finite. The case in which the number of representatives is infinite is thus less trivial, since we need to refine our classification criteria and go beyond QG-bordisms. At this point, several possibilities open up and it is not clear yet what the correct strategy would be, as one could classify null-QG-bordism representatives up to e.g. diffeomorphisms, QG-preserving homeomorphisms, QG-symmetries, or other deformations. In other words, one has to define a refined equivalence relation, which is not easy to identify at this stage. To make progress, one could concentrate on a subpart of the problem. A precise statement in the literature is the one on the finiteness of Calabi–Yau manifolds mentioned above, which can be rephrased as the existence of bounds on the corresponding Hodge numbers. In this case, one is thus led to state that the number of allowed QG-backgrounds is finite up to deformations, which is a refinement with respect to QG-bordisms. Arguably, proving this statement (and its extensions) could be as challenging as proving its counterpart in the finiteness conjecture. To illustrate the non-trivial nature of the problem we may consider the simple example $\Omega_2^{\text{SO}} = 0$. This group is trivial, but there are infinitely many (orientation-preserving) diffeomorphism classes of representatives of $[\varnothing]$, because oriented closed surfaces are classified by their genus. This clearly illustrates that one can have in general different ways of refining the equivalence classes of the null-QG-bordism representatives, in this case for example by diffeomorphism classes or by orientation-preserving diffeomorphism classes. In summary, to rephrase the finiteness conjecture in terms of the cobordism conjecture, one is led to a statement of the following type: the number of representatives of the single element of $\Omega_k^{\text{QG}}$ is finite, up to an appropriate equivalence relation.

## Acknowledgements

We are particularly grateful to D. Fiorenza and M. Montero for illuminating discussions and valuable comments on an earlier version of the manuscript. We also thank R. Blumenhagen, H. Skarke, and F. Thuillier for useful exchanges at various stages of this project. N. Carqueville is supported by the DFG Heisenberg Programme. The work of N. Cribiori is supported by the Alexander von Humboldt Foundation and was supported by an FWF grant with the number P 30265 at the initial stage of this project.

# A  *NS*-branes and higher brane structures

In this appendix, we briefly review *NS*-branes, before discussing their possible relations to "higher brane structures" introduced in [46] and appearing in the Whitehead tower in Section 3. The $NS_5$-brane is a well-known extended object in string theory, $S$-dual to the type IIB $D_5$-brane and appearing in the following chain of T-dualities:

$$NS_5\text{-brane} \quad \longleftrightarrow \quad KK\text{-monopole} \quad \longleftrightarrow \quad Q\text{-brane}. \tag{A.1}$$

It is also a solution of 10-dimensional supergravity, and it serves as the magnetic source to an $H$-flux. Its worldvolume is 6-dimensional, so it is a codimension-4 object in ten dimensions. The meaning of the chain (A.1) is the following. Considering an $NS_5$-brane and smearing it along one of its transverse dimensions and T-dualising along that direction, one obtains the Kaluza–Klein monopole ($KKm$). As such, the latter is a codimension-3 object in ten dimensions. Its space dimensions are however special: they split into five dimensions plus one isometry direction,[16] where the latter is fibred over the transverse dimensions. The $KKm$ is also a supergravity solution, and it admits no $H$-flux. Rather, it is a purely geometric background, and it can be viewed as magnetically sourcing "geometric flux", i.e. metric curvature, as discussed in Section 4.3. A further smearing of a transverse dimension followed by a T-duality leads to a last background, which turns out to be a "non-geometry", more precisely a "T-fold". On the latter, tensor fields do not only glue on overlapping patches by diffeomorphisms as on a differentiable manifold, but also via T-duality transformations (see e.g. [59, Sect. 4.2.2]); more generally, a non-geometry refers to a background where the gluing is made via stringy symmetries. The resulting codimension-2 object, along five dimensions plus two isometry directions, is a first instance of "exotic branes", which are non-geometric objects obtained by T- or S-dualities from standard branes of string theory; see e.g. [86, Fig. 1] for an illustration of such brane duality webs. This non-geometric brane was first identified and named $5_2^2$-brane in [87, 88]. It was then realised that exotic branes could source non-geometric fluxes, and the $5_2^2$-brane was identified as the magnetic source of the $Q$-flux [59, 89], hence renamed as the $Q$-brane. We refer to the review [66] for proper references, and to e.g. [59, Sect. 3.2] for explicit background fields and (magnetically) sourced Bianchi identities.

The paper [46] introduced the "higher brane structures" $O\langle 9 \rangle$ and $O\langle 10 \rangle$, that appear in the Whitehead tower discussed in Section 3. One may wonder whether stringy objects can be associated to those two structures, as is the case for some of the other structures in the tower. We propose here the following interpretation: we dub $O\langle 9 \rangle$ a "Kaluza–Klein monopole" structure, and $O\langle 10 \rangle$ a "$Q$-brane" structure, referring to the stringy objects just reviewed.[17] A simple justification comes from the following observation. Given an $O\langle n+1 \rangle$-structure, one can find a corresponding stringy object with an $(n-1)$-dimensional worldvolume; equivalently, $11 - n$ corresponds to its codimension in ten dimensions.[18] This is true for String, Fivebrane and Ninebrane structures. With the above proposal, it would then also be true for the Kaluza–Klein monopole and the $Q$-brane, which are as well NSNS objects. Another point in favour of

---

[16]As is common in the context of T-duality, the term "isometry direction" refers to a dimension associated to a coordinate $x$, on which the background fields do not depend.

[17]The names "2-orientation" and "2-spin" introduced in [46] for $O\langle 9 \rangle$ and $O\langle 10 \rangle$ come from Bott periodicity: for instance, $9 = 1 + 8$ and $O\langle 1 \rangle = SO$. This has a priori no relation to stringy objects.

[18]Notice that it can happen that $O\langle n+1 \rangle = O\langle n+2 \rangle = \ldots$. In this case we are referring to $O\langle n+1 \rangle$.

the proposal is that $\pi_{10}(\mathrm{O}) = 0$, thus $\mathrm{O}\langle 10 \rangle = \mathrm{O}\langle 11 \rangle$, so one would not introduce yet another object for $\mathrm{O}\langle 11 \rangle$. This is reminiscent of what happens when using conventional Buscher T-duality rules: due to the absence of more isometry directions, one stops at the $Q$-brane and cannot reach through T-duality a further hypothetical object (sometimes referred to as the non-geometric $R$-brane).

To put this proposal on solid grounds, one should match the obstructions to these structures with the anomalies cured by the corresponding stringy object. Consider a structure carrying the name of a certain object, and take this object as an electric source. One should then study the Bianchi identity for the corresponding *electric* field strength [44, 46]. One reads from these $\alpha'$-corrected Bianchi identities the characteristic classes corresponding to topological obstructions to that structure. For example, the heterotic Bianchi identity for the $H$-flux contains $p_1(\omega)$, see [44, Eq. (14)]. In this case the electric source is a string, and $\frac{1}{2}p_1$ serves as an obstruction to the string structure. The same goes for the Fivebrane structure and $\mathrm{d}H_7$, see [44, Eq. (15)]. What can be said for the Kaluza–Klein monopole and $Q$-brane? The Bianchi identity for them as *magnetic* sources were identified e.g. in [59] (see also (4.20)), but one would need the identities for them as *electric* sources, and we do not know them.

Alternatively, we may consider the magnetic Bianchi identities of the electromagnetic dual objects. The electromagnetic counterpart of the Kaluza–Klein monopole is a specific $pp$-wave, its worldvolume is 1-dimensional. We would need the Bianchi identity magnetically sourced by that wave, to hopefully read the obstruction to an $\mathrm{O}\langle 9 \rangle$-structure.[19] That Bianchi identity might be read from [73], though we may need the $\alpha'$-corrected version. Regarding $\mathrm{O}\langle 10 \rangle$, one may consider the electromagnetic counterpart of a codimension-2 object, which is more difficult to interpret. Interestingly, this difficulty occurs precisely for a non-geometric brane. We also note that "gauge fields" associated to non-geometric branes are sometimes $p$-vectors rather than $p$-forms. Such tensors and their Bianchi identities may then provide alternative characterisations of the relevant obstructions. On the mathematical side, we also note that the obstructions to $\mathrm{O}\langle 9 \rangle$- and $\mathrm{O}\langle 10 \rangle$-structures have been identified in [46] but their actual calculation is non-trivial.

---

[19]Along the same lines, the Bianchi identity magnetically sourced by a Kaluza–Klein monopole should provide the obstruction to $n = 2$, i.e. the $\mathrm{O}\langle 3 \rangle$-structure which is the Spin structure. This Bianchi identity was identified in [59] to be the Riemann Bianchi identity (see also (4.20)). This implies that $\alpha'$-corrections to this Bianchi identity should carry a Stiefel–Whitney class number, e.g. $w_3$. It would be interesting to verify this claim.

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
