# Peer review of "Looking for structure in the cobordism conjecture"

_SciPost Physics_

## Round 1 · Referee Report · Anonymous (Referee 1) · 2022-6-2

Report

The authors consider several aspects of the cobordism conjecture, a very relevant conjecture in the Swampland program and that has concentrated much activity in the recent years. In this paper, the authors focus in its mathematical formulation.

The manuscript is well-written and self-contained. For this purpose, the authors review several results and mathematical constructions. Section 2 reviews the definition of bordism groups, as well as the cobordism and no global symmetries conjecture and their relation. Section 3 contain a review of the Whitehead tower construction of the orthogonal group, as well as the physical interpretation of some obstructions that appear in it. Section 4 reviews how the cobordism conjecture can be used to predict the presence of several defects.

They use the reviewed material to make several proposals: In section 3, the Whitehead tower construction is proposed as an organizing principle for the different structures that may enter in the cobordism groups. In particular, they argue that it can be useful to identify which extra ingredients may help to trivialize some cobordism groups, as required by the cobordism conjecture. In section 4, they propose a formal definition for the cobordism groups including p-form fluxes and the defects that must be added to absorb them. It is also argued that KK monopoles may serve as defects trivializing certain cobordism group in string theory. In section 5, the authors discuss the gravity decoupling limit of the conjecture. Finally, in the appendix, they propose a physical interpretation for a couple of floors in the Whitehead tower of the orthogonal group that were missing it.

I have a couple of questions/suggestions that I would like the authors to address:

$\bullet$ About the procedure of replacing the defect by a sphere with certain units of flux. As nicely pointed out by the authors in footnote 8, this can be regarded as an effective description. As such, do the authors expect that it could loose some information for the cobordism conjecture with respect to the description using defects? For example, a remarkable feature of the cobordism conjecture is that it predicts that QG should make sense in certain singular manifolds (with the singularity describing the defect). This happens for instance when the cobordism trivializing some class involves orbifolds, which are known to be consistent in string theory.

$\bullet$ I would ask the authors to further explain the reasoning leading to (4.15). As stated above (4.13), the bordism between $(M,n_M)$ and $(N,n_N)$ with defects can be replaced by the bordism between $(M,n_M)\sqcup-(N,n_N)$ and, in general, n copies of $(S^{k},n_j)$ without defects. Therefore, the classes in the cobordism group including the defect include all the classes in the cobordism group without defects that can be obtained by multiplication by the class of $(S^{k},n_j)$. Is this the identification of pairs that the authors argue to give the quotient in (4.15)?

$\bullet$ Along the whole manuscript, it seems to be implied that the QG structure in $\Omega_{k}^{QG}$ is unique. Should this really be the case? A priori one could expect to find different ones. All of them should be trivial, but would correspond to different structures in the mathematical sense. This does not mean that they could not be "dual", as for example the authors argue that happens for some cobordism groups in the last paragraph of page 29.

I would like to give a further suggestions that, in my opinion, would improve the quality of a revised version of the manuscript. Section 2 presents a very detailed introduction to the notion of cobordism groups and their refinements.
However, some technical concepts such as the classifying space or stable groups may not be familiar for part of the target audience. For this reason, I think giving some more physical motivation and intuition could help understanding them better. For example, one could motivate why physicists are interested in adding $G$-bundles, as part of the audience may find it non-trivial. In addition, it would be helpful to give some intuition about the meaning of adding these into the definition of cobordism groups before entering into the technical details. The same applies to the concept of tangential structures. In any case, these modifications are not crucial and I therefore leave it to the authors to take them into account or not.

---

## Round 1 · Referee Report · Anonymous (Referee 2) · 2022-7-12

Strengths

1- The paper is clearly written.
2- The subject explored is interesting.

Report

The authors of this paper explore the cobordism conjecture, which asserts that the (yet unknown) right cobordism theory $\Omega^{\text{QG}}$ in quantum gravity trivialises. One of the main goals is to understand if the Whitehead towers helps in identifying which is the right cobordism theory. This is a very reasonable guess, since

1. The Whitehead tower involves "killing off" lower dimensional homotopy groups, and homotopy is closely related to bordism.

2. Familiar structures in string theory, like string and M5 structures, arise naturally on the Whitehead tower.

The authors explore this direction. Eventually the analysis does not clarify much the possible implications of the Whitehead tower for $\Omega^{\text{QG}}$ but this was an interesting direction to explore, and the analysis is clear, so these are good results.

There are also some results on fluxes and defects, which clarify various aspects of the proposal, and a discussion of gravity decoupling.

Overall I find the paper interesting and well written, but I would like the authors to address a couple of points (in addition to the points raised by the other referee):

1. I think that the review of gauging in section 2.3.2 could have been written more clearly. The authors seem to be following [39], but made some changes that make the discussion hard to understand. For instance, although I am sure that the authors know this, it would be good to avoid statements like "by introducing an additional source term in the action ... we gauge the global symmetry". It is in general good to distinguish the act of coupling to background fields and the gauging itself (that is, the sum over backgrounds). Also $j_{k+1}$ is never defined, and it is unclear if it has anything to do with $j_k$. A related point, is that it is not clearly identified what is background and what is current: ideally one would introduce a background $B_{d-k}$ for the global symmetry, coupled to the current $j_k$. Gauging is then integrating over $B_{d-k}$ (with a kinetic term $H_{d-k+1}\wedge \star H_{d-k+1}$ in the continuous case.

2. This is probably a small typo, but $\frac{1}{2}p_2(TM)$ at the top of page 18 should be $\frac{1}{2}p_1(TM)$ instead, I believe.

3. I am a little puzzled about the definition of the "KK monopole problem in section 4.3". This object is purely gravitational, so it is already present in smooth/spin bordism. Why does it need predicting, any more than any other geometry (local CY threefold singularities, say)? It would be good if the authors elaborated on this, I don't think I understand the motivation for this section.

---

## Editorial Decision

resubmitted